# Variational Potential Flow: A Novel Probabilistic Framework for Energy-Based Generative Modelling

## Abstract

Energy based models (EBMs) are appealing for their generality and simplicity in data likelihood modeling, but have conventionally been difficult to train due to the unstable and time-consuming implicit MCMC sampling during contrastive divergence training. In this paper, we present a novel energy-based generative framework, Variational Potential Flow (VAPO), that entirely dispenses with implicit MCMC sampling and does not rely on complementary latent models or cooperative training. The VAPO framework aims to learn a potential energy function whose gradient (flow) guides the prior samples, so that their density evolution closely follows an approximate data likelihood homotopy. An energy loss function is then formulated to minimize the Kullback-Leibler divergence between density evolution of the flow-driven prior and the data likelihood homotopy. Images can be generated after training the potential energy, by initializing the samples from Gaussian prior and solving the SDE governing the potential flow. Experiment results show that the proposed VAPO framework is capable of generating realistic images on various image datasets. In particular, our proposed framework achieves competitive FID scores for unconditional image generation on the CIFAR-10 and CelebA datasets.

## 1 Introduction

In recent years, deep generative modeling has garnered significant attention for unsupervised learning of complex, high-dimensional data distributions Bond-Taylor et al. (2022). In particular, probabilistic generative models such as variational autoencoders Kingma & Welling (2022), normalizing flows Rezende & Mohamed (2015), score matching or diffusion models Song et al. (2021); Kim et al. (2021); Karras et al. (2022). Poisson flow Xu et al. (2022; 2023), and energy-based models (EBMs) Du & Mordatch (2019); Grathwohl et al. (2020a) aim to maximize the likelihood (probability density) underlying the data. By design, these probabilistic frameworks enhance training stability, accelerate model convergence, and reduce mode collapse compared to generative adversarial networks Srivastava et al. (2017), albeit at the cost of a slow sampling procedure and poor model scalability Grathwohl et al. (2020b). Among these frameworks, EBMs have emerged as a flexible and expressive class of probabilistic generative models Du & Mordatch (2019); Grathwohl et al. (2020b); Du et al. (2021); Nijkamp et al. (2019); Gao et al. (2020; 2021); Zhu et al. (2024); Grathwohl et al. (2020a); Yang et al. (2023). EBMs model high-dimensional data space with a network-parameterized energy potential function that assigns data regions with energy that is directly (or inversely) proportional to the unnormalized data likelihood Song & Kingma (2021). This provides a natural interpretation of the network model in the form of an energy landscape, endowing EBMs with inherent interpretability.

Deep EBMs are particularly appealing since they impose no restrictions on the network architecture, potentially resulting in high expressiveness Bond-Taylor et al. (2022). Moreover, they are more robust and generalize well to out-of-distribution samples Du & Mordatch (2019); Grathwohl et al. (2020a) as regions with high probability under the model but low probability under the data distribution are explicitly penalized during training. Additionally, EBMs, which trace back to Boltzmann machines Hinton (2002), have strong ties to physics models and can thus borrow insights and techniques from statistical physics for their development and analysis Feinauer & Lucibello (2021). On these grounds, EBMs have been applied across a diverse array of applications apart from image modelling,

including text generation Deng et al. (2020); Yu et al. (2022), point cloud synthesis Xie et al. (2021a), scene graph generation Suhail et al. (2021), anomaly detection Vu et al. (2017); Yoon et al. (2023), earth observation Castillo-Navarro et al. (2022), robot learning Du et al. (2020); Sodhi et al. (2022), trajectory prediction Pang et al. (2021); Wang et al. (2023), and molecular design Liu et al. (2021); Sun et al. (2021).

Despite a number of desirable properties, deep EBMs require implicit Langevin Markov Chain Monte Carlo (MCMC) sampling during the contrastive divergence training. MCMC sampling in a high-dimensional setting, however, has shown to be challenging due to poor mode mixing and excessively long mixing time Bond-Taylor et al. (2022); Du & Mordatch (2019); Nijkamp et al. (2019); Gao et al. (2020); Grathwohl et al. (2020a); Nijkamp et al. (2022). As result, energy potential functions learned with non-convergent MCMC do not have valid steady-states, in the sense that MCMC samples can differ greatly from data samples Grathwohl et al. (2020b). Current deep EBMs are thus plagued by high variance training and high computational complexity due to MCMC sampling. In view of this, recent works have explored learning complementary latent model to amortize away the challenging MCMC sampling Xiao et al. (2021); Grathwohl et al. (2021); Hill et al. (2022); Pang et al. (2020); Yu et al. (2023), or cooperative learning where model-generated samples serve as initial points for subsequent MCMC revision in the latent space Xie et al. (2020); Cui & Han (2023). While such approaches alleviate the burden of MCMC sampling, it comes at the expense of the inherent flexibility and composability of EBMs Du et al. (2021).

In this paper, we introduce Variational Potential Flow (VAPO), a novel energy-based generative framework that eliminates the need for implicit MCMC sampling and complementary models. At the core of VAPO lies the construction of a homotopy (smooth path) that bridges the prior distribution with the data likelihood. Subsequently, a potential flow with model-parameterized potential energy function is designed to guide the evolution of prior sample densities along this approximate data likelihood homotopy. Applying a variational approach to this path-matching strategy ultimately yields a probabilistic Poisson's equation, where the weak solution corresponds to minimizing the energy loss function of our proposed VAPO.

Our contributions are summarized as follows:

- We introduce VAPO, a novel energy-based generative framework that entirely dispenses with the unstable and inefficient implicit MCMC sampling. Our proposed framework learns a potential energy function whose gradient (flow) guides the prior samples, ensuring that their density evolution path closely follows the approximate data likelihood homotopy.

- We derive an energy loss function for VAPO by constructing a variational formulation of the intractable homotopy path-matching problem. Solving this energy loss objective is equivalent to minimizing the Kullback-Leibler divergence between density evolution of the flow-driven prior and the approximate data likelihood homotopy.

- To assess the effectiveness of our proposed VAPO for image generation, we conduct experiments on the CIFAR-10 and CelebA datasets and benchmark the performances against state-of-the-art generative models. Our proposed framework achieves competitive FID scores for unconditional image generation.

## 2 BACKGROUND AND RELATED WORKS

In this section, we provide an overview of EBMs, particle flow, and the deep Ritz method, collectively forming the cornerstone of the proposed VAPO framework. Additionally, we discuss more related works on diffusion, flow-based, and energy-based generative models in Appendix A.

### 2.1 ENERGY-BASED MODELS

Denote $\bar{x} \in \Omega \subseteq \mathbb{R}^n$ as the training data, EBMs approximate the data likelihood $p_{\text{data}}(\bar{x})$ via defining a Boltzmann distribution $p_\theta(x) = \frac{e^{\Phi_\theta(x)}}{Z(\theta)}$ where $\Phi_\theta$ is an energy model and $Z(\theta) = \int_\Omega e^{\Phi_\theta(x)} \, dx$ is the normalizing partition function. Given that this partition function is analytically intractable for high-dimensional data, EBMs perform the maximum likelihood estimation (MLE) by minimizing the negative log likelihood loss $\mathcal{L}_{\text{MLE}}(\theta) = \mathbb{E}_{p_{\text{data}}(\bar{x})}[\log p_\theta(\bar{x})]$ and approximate its gradient via the

contrastive divergence Hinton (2002):

$$\nabla_\theta \mathcal{L}_{\text{MLE}} = \mathbb{E}_{p_{\text{data}}(\bar{x})}\big[\nabla_\theta \Phi_\theta(\bar{x})\big] - \mathbb{E}_{p_\theta(x)}\big[\nabla_\theta \Phi_\theta(x)\big] \tag{1}$$

However, EBMs are computationally intensive due to the implicit MCMC generating procedure, required for generating negative samples $x \in \Omega \sim p_\theta(x)$ for gradient computation (1) during training.

## 2.2 PARTICLE FLOW

Particle flow, initially introduced by the series of papers Daum & Huang (2007), is a class of nonlinear Bayesian filtering (sequential inference) methods that aim to approximate the posterior distribution $p(x_t|\bar{x}_{0:t})$ of the state of system given the observations. While particle flow methods are closely related to normalizing flows Rezende & Mohamed (2015) and neural ordinary differential equations (ODEs) Chen et al. (2018), these latter frameworks do not explicitly accommodate a Bayes update. In particular, particle flow performs the Bayes update $p(x_t|\bar{x}_{0:t}) \propto p(x_t|\bar{x}_{0:t-1})\, p(\bar{x}_t|x_t, \bar{x}_{0:t-1})$ by subjecting prior samples $x_t \sim p(x_t|\bar{x}_{0:t-1})$ to a series of infinitesimal transformations through an ODE $\frac{dx}{d\tau} = v(x, \tau)$ parameterized by a flow velocity (field) function $v(x, \tau)$, in a pseudo-time interval $\tau \in [0, 1]$ in between sampling time steps. The flow velocity is designed such that the driven Kolmogorov forward path evolution (Fokker–Planck dynamics, see (12)) of the sample particles, coincides with a data log-homotopy (smooth path) that inherently perform the Bayes update. Despite its efficacy in time-series inference Pal et al. (2021); Chen et al. (2019b); Yang et al. (2014) and resilience to the curse of dimensionality Surace et al. (2019), particle flow has yet to be explored in generative modelling for high-dimensional data.

## 2.3 DEEP RITZ METHOD

The deep Ritz method is a deep learning-based variational numerical approach, originally proposed in E & Yu (2018), for solving scalar elliptic partial differential equations (PDEs) in high dimensions. Consider the following Poisson's equation, fundamental to many physical models:

$$\Delta u(x) = f(x),\ x \in \Omega \quad \text{subject to} \quad u(x) = 0,\ x \in \partial\Omega \tag{2}$$

where $\Delta$ is the Laplace operator, and $\partial\Omega$ denotes the boundary of $\Omega$. For a Sobolev function $u \in \mathcal{H}_0^1(\Omega)$ (see Proposition 2 for definition) and square-integrable $f \in L^2(\Omega)$, the variational principle ensures that a weak solution of the Euler-Lagrange boundary value equation (2) is equivalent to the variational problem of minimizing the Dirichlet energy Müller & Zeinhofer (2019), as follows:

$$u^* = \arg\min_v \int_\Omega \left( \frac{1}{2}\|\nabla v(x)\|^2 - f(x)v(x) \right) dx \tag{3}$$

where $\nabla$ denotes the Del operator (gradient). In particular, the deep Ritz method parameterizes the trial energy function $v$ using neural networks, and performs the optimization (3) via stochastic gradient descent. Due to its versatility and effectiveness in handling high-dimensional PDE systems, the deep Ritz method is predominantly applied for finite element analysis Liu et al. (2023a). In Olmez et al. (2020), the deep Ritz method is used to solve the probabilistic Poisson's equation resulting from the feedback particle filter Yang et al. (2013). Nonetheless, the method has not been explored for generative modelling.

## 3 VARIATIONAL ENERGY-BASED POTENTIAL FLOW

In this section, we introduce a novel generative modelling framework, Variational Energy-Based Potential Flow (VAPO), drawing inspiration from both particle flow and the calculus of variations. First, we establish a homotopy that transforms a prior to the data likelihood and derive the evolution of the prior in time. Then, we design an energy-generated potential flow and a weighted Poisson's equation that aligns the evolving density distribution of transported particles with the homotopy-driven prior. Subsequently, we formulate a variational loss function where its optimization with respect to the flow-generating potential energy is equivalent to solving the Poisson's equation. Finally, we describe the model architecture that is used to parameterize the potential energy function and the backward SDE integration for generative sampling.

### 3.1 Bridging Prior and Data Likelihood: Log-Homotopy Transformation

Let $\bar{x} \in \Omega$ denote the training data, $p_{\text{data}}(\bar{x})$ be the data likelihood, $x \in \Omega$ denote the approximate data samples. To achieve generative modelling, our objective is to closely approximate the training data $\bar{x}$ with the data samples $x$. On this account, we define a conditional data likelihood $p(\bar{x}|x) = \mathcal{N}(\bar{x}; x, \Pi)$ with isotropic Gaussian noise with covariance $\Pi = \text{diag}(\sigma^2)$ and standard deviation $\sigma \in \Omega$. This is equivalent to considering a state space model $x = \bar{x} + \nu$, where $\nu \in \Omega \sim \mathcal{N}(\nu; 0, \Pi)$. Here, we set a small $\sigma$ so that $x$ closely resembles the training data $\bar{x}$.

Consider a conditional (data-conditioned) density function $\rho : \Omega^2 \times [0,1] \to \mathbb{R}$, as follows:

$$\rho(x; \bar{x}, t) = \frac{e^{f(x; \bar{x}, t)}}{\int_\Omega e^{f(x; \bar{x}, t)} \, dx} \tag{4}$$

where $f : \Omega^2 \times [0,1] \to \mathbb{R}$ is a log-linear function:

$$f(x; \bar{x}, t) = \log q(x) + t \log p(\bar{x}|x) \tag{5}$$

parameterized by the auxiliary time variable $t \in [0,1]$, and we let $q(x) = \mathcal{N}(x; 0, \Lambda)$ be a isotropic Gaussian prior density with covariance $\Lambda = \text{diag}(\omega^2)$ and standard deviation $\omega \in \Omega$. Here, $\text{diag}(\cdot)$ denotes the diagonal function. By construction, we have $\rho(x; \bar{x}, 0) = q(x)$ at $t = 0$, and $\rho(x; \bar{x}, 1) = p(x|\bar{x})$ at $t = 1$ since we have

$$\rho(x; \bar{x}, 1) = \frac{e^{f(x; \bar{x}, 1)}}{\int_\Omega e^{f(x; \bar{x}, 1)} \, dx} = \frac{p(\bar{x}|x) \, q(x)}{\int_\Omega p(\bar{x}|x) \, q(x) \, dx} = \frac{p(\bar{x}, x)}{\int_\Omega p(\bar{x}, x) \, dx} = p(x|\bar{x}) \tag{6}$$

where we have used the fact that $p_{\text{data}}(\bar{x}) = \int_\Omega e^{f(x; \bar{x}, 1)} \, dx = \int_\Omega p(x, \bar{x}) \, dx$. Therefore, the conditional density function $\rho(x; \bar{x}, t)$ here (4) essentially represents a density homotopy between the prior $q(x)$ and the posterior $p(x|\bar{x})$.

In particular, the density function $\rho(x; \bar{x}, t)$ also defines a conditional (data-conditioned) homotopy between the prior $q(x)$ and the exact posterior $p(x|\bar{x})$, the latter of which gives a maximum a posteriori (Bayesian) estimate of the approximate data samples after observing true training data.

To obtain an estimate of the intractable data likelihood for generative sampling, we then consider a (approximate) data likelihood homotopy $\bar{\rho} : \Omega \times [0,1] \to \mathbb{R}$ as follows:

$$\bar{\rho}(x; t) = \int_\Omega p_{\text{data}}(\bar{x}) \, \rho(x; \bar{x}, t) \, d\bar{x} \tag{7}$$

Considering this, it remains that $\bar{\rho}(x; 0) = q(x)$ at $t = 0$. Furthermore, given that we have $\bar{\rho}(x; 1) = \int_\Omega p_{\text{data}}(\bar{x}) \, p(x|\bar{x}) \, d\bar{x} = \bar{p}(x)$ at $t = 1$, the data likelihood homotopy $\bar{\rho}(x; t)$ here inherently performs a kernel density approximation of the true data likelihood, using the normalized kernel $p(x|\bar{x})$ obtained from the conditional homotopy $\rho(x; \bar{x}, 1)$ at $t = 1$. Therefore, the approximate data likelihood $\bar{p}(x)$ acts as a continuous interpolation of the data likelihood $p_{\text{data}}(x)$, represented by Dirac delta function $\delta(x - \bar{x})$ centered on the discrete training data $\bar{x}$.

Nevertheless, the conditional homotopy (6) is intractable due to the normalizing constant in the denominator. This intractability rules out a close-form solution of the data likelihood homotopy (7), thus it is not possible to sample directly from the data likelihood estimate. Taking this into account, we introduce the potential flow method in the following section, where we model the evolution of the prior samples (particles) instead, such that their distribution adheres to the data likelihood homotopy.

### 3.2 Modelling Potential Flow in a Homotopy Landscape

Our aim is to model the flow of the prior particles in order for their distribution to follow the data likelihood homotopy and converge to the data likelihood. To accomplish this, we first derive the evolution of the latent prior density with respect to time in the following proposition.

**Proposition 1.** *Consider the data likelihood homotopy $\bar{\rho}(x; t)$ in (7) with Gaussian conditional data likelihood $p(\bar{x}|x) = \mathcal{N}(\bar{x}; x, \Pi)$. Then, its evolution in time $t \in [0,1]$ is given by the following PDE:*

$$\frac{\partial \bar{\rho}(x; t)}{\partial t} = -\frac{1}{2} \, \mathbb{E}_{p_{data}(\bar{x})} \left[ \rho(x; \bar{x}, t) \left( \gamma(x, \bar{x}) - \bar{\gamma}(x, \bar{x}) \right) \right] \tag{8}$$

*where*

$$\gamma(x, \bar{x}) = (x - \bar{x})^T \Pi^{-1} (x - \bar{x}) \qquad (9)$$

*is the innovation (weighted residual sum of squares) term in the conditional data likelihood, and $\bar{\gamma}(x, \bar{x}) = \mathbb{E}_{\rho(x;\bar{x},t)}[\gamma(x, \bar{x})]$ denotes the expectation of the innovation with respect to the conditional homotopy and on the latent variables.*

*Proof.* Refer to Appendix B.1. □

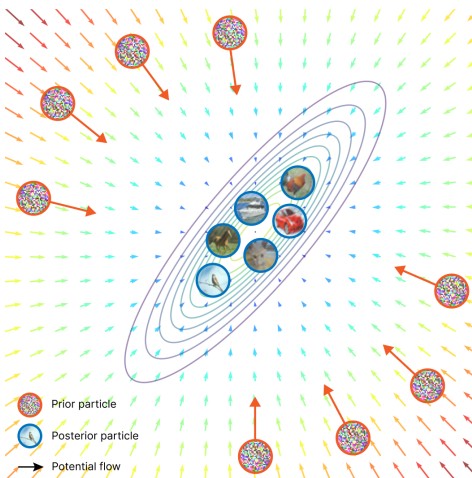

Figure 1: A planar visualization of the potential-generated field (represented by coloured arrows) that transports the prior particles towards the approximate data likelihood (represented by the blue contour).

Our proposed potential flow method involves subjecting the latent prior samples to a potential-generated velocity field, such that the flow trajectories of these sample particles $x(t)$ within the interval $t \in [0, 1]$ are governed by the following stochastic differential equation (SDE):

$$dx(t) = \nabla\Phi(x(t), t) \, dt + \beta(t) \, dW_t \qquad (10)$$

which is closely related to the stochastic Langevin dynamic, typically considered in MCMC-based EBMs. Here, $\Phi : \Omega \times [0, 1] \to \mathbb{R}$ is a scalar potential energy function, $\beta : [0, 1] \to \mathbb{R}$ is the diffusion coefficient and $W_t \in \mathbb{R}^n$ denote a standard Wiener process. Therefore, $\nabla\Phi \in \Omega$ is the velocity vector field generated by the potential energy, and $\nabla$ denotes the Del operator (gradient) with respect to the data samples $x$. Henceforth, we omit the time variable $t$ in $\Phi(\cdot, t)$ and $x(t)$ for readability.

Considering a potential flow of the form (10), a direct consequence is that the prior density $\rho^\Phi$ of the flow-driven prior samples evolves according to a Fokker–Planck (Kolmogorov forward) equation Gardiner (2009), as follows:

$$\frac{\partial \rho^\Phi(x; t)}{\partial t} = -\nabla \cdot \left( \rho^\Phi(x; t) \, \nabla\Phi(x) \right) + \frac{\beta(t)^2}{2} \Delta\rho^\Phi(x; t) \qquad (11)$$

where $\nabla\cdot$ denotes the divergence operator. In particular, this Fokker–Planck equation is known to possess an equilibrium (steady state) solution in the form of the Boltzmann distribution $\rho^\Phi \propto e^{\Phi(x)}$, thereby establishing a connection to EBM. Moreover, (11) can be equivalently expressed as a continuity (transport) equation, as follows:

$$\frac{\partial \rho^\Phi(x; t)}{\partial t} = -\nabla \cdot \left( \rho^\Phi(x; t) \, \nabla\Phi^*(x) \right) \qquad (12)$$

which corresponds to an ODE $\frac{dx}{dt} = \nabla\Phi^*(x)$ with a field-generating potential of the form $\Phi^*(x) = \Phi(x) - \frac{\beta(t)^2}{2} \log \rho^\Phi(x; t)$. This continuity representation of the Fokker–Planck equation (12) will be instrumental in our subsequent formulation of the energy loss function.

The goal of our proposed framework is to model the potential energy function in the potential flow (10), such that the progression of the prior density subject to potential flow emulates the evolution of the data likelihood homotopy. In particular, we seek to solve the problem of minimizing the Kullback-Leibler divergence in the following proposition.

**Proposition 2.** *Consider a potential flow of the form (10) and given that $\Phi \in \mathcal{H}_0^1(\Omega, \bar{\rho})$, where $\mathcal{H}_0^n$ denotes the (Sobolev) space of $n$-times differentiable functions that are compactly supported, and square-integrable with respect to data likelihood homotopy $\bar{\rho}(x; t)$.*

*The problem of solving for the optimal potential energy function $\Phi(x)$ that satisfies the following probabilistic (density-weighted) Poisson's equation:*

$$\nabla \cdot \left( \bar{\rho}(x; t) \, \nabla\Phi^*(x) \right) = \frac{1}{2} \, \mathbb{E}_{p_{data}(\bar{x})} \left[ \rho(x; \bar{x}, t) \left( \gamma(x, \bar{x}) - \bar{\gamma}(x, \bar{x}) \right) \right] \qquad (13)$$

where $\Phi^*(x) = \Phi(x) - \frac{\beta(t)^2}{2} \log \bar{\rho}(x; t)$, *is then equivalent to minimizing the Kullback-Leibler divergence* $\mathcal{D}_{\mathrm{KL}} \left[ \rho^\Phi(x; t) \parallel \bar{\rho}(x; t) \right]$ *between the flow-driven prior* $\rho^\Phi(x; t)$ *and the data likelihood homotopy* $\bar{\rho}(x; t)$.

*Proof.* Refer to Appendix B.2. □

In hindsight, the left-hand side of the probabilistic Poisson's equation resembles the evolution of the flow-driven prior given by the Fokker-Plank equation (12). In addition, the right-hand side resembles the evolution of data likelihood homotopy given by PDE (8), with the conditional homotopy $\rho(x; \bar{x}, t)$ replaced by flow-driven prior $\rho^\Phi(x; t)$. Therefore, Proposition 2 is an attempt to perform the density homotopy path-matching $\rho^\Phi(x; t) \equiv \bar{\rho}(x; t)$ via solving the Poisson's equation (13). In this context, the flow-driven prior $\rho^\Phi$ can be interpreted as an approximate likelihood density. It is also essential to emphasize that Proposition 2 plays a crucial role in density path-matching, as it enables us to solve (13) which is expressed solely in terms of $\rho(x; \bar{x}, t)$ and $\bar{\rho}(x; t)$, both of which permit direct sampling. This approach bypasses the intractable Kullback-Leibler divergence (KLD) $\mathcal{D}_{\mathrm{KL}} \left[ \rho^\Phi(x; t) | \bar{\rho}(x; t) \right]$, given that direct sampling from $\rho^\Phi(x; t)$ is not feasible.

Nonetheless, obtaining an explicit solution to (13) is challenging in a high-dimensional setting. Numerical methods that approximate the solution often do not scale well with the data dimension. For example, the Galerkin approximation requires a selection of the basis functions, which becomes non-trivial when the dimensionality is high Yang et al. (2016). The diffusion map-based algorithm, on the other hand, requires a large number of particles, which grows exponentially with respect to the dimensionality, in order to achieve error convergence Taghvaei et al. (2020). Taking this into consideration, we propose an energy loss function in the following section, where we cast the Poisson's equation as a variational problem compatible with stochastic gradient descent.

### 3.3 VARIATIONAL ENERGY LOSS FUNCTION FORMULATION: DEEP RITZ APPROACH

In this section, we introduce an energy method which presents a variational formulation of the probabilistic Poisson's equation. Given that the aim is to minimize the divergence between the data likelihood homotopy and the flow-driven prior and directly solving the probabilistic Poisson's equation is difficult, we first consider a weak formulation of (13) as follows:

$$\int_\Omega \left( \frac{1}{2} \, \mathbb{E}_{p_{\mathrm{data}}(\bar{x})} \left[ \rho(x; \bar{x}, t) \left( \gamma(x, \bar{x}) - \bar{\gamma}(x, \bar{x}) \right) \right] - \nabla \cdot \left( \bar{\rho}(x; t) \, \nabla \Phi^*(x) \right) \right) \Psi(x) \, dx = 0 \quad (14)$$

where the equation must hold for all differentiable trial functions $\Psi$. In the following proposition, we introduce an energy loss objective that is equivalent to solving this weak formulation of the probabilistic Poisson's equation.

**Proposition 3.** *The variational problem of minimizing the following loss function:*

$$\mathcal{L}(\Phi^*; t) = \frac{1}{2} \operatorname{Cov}_{\rho(x; \bar{x}, t) \, p_{data}(\bar{x})} \left[ \Phi^*(x), \gamma(x, \bar{x}) \right] + \frac{1}{2} \, \mathbb{E}_{\bar{\rho}(x; t)} \left[ \left\| \nabla \Phi^*(x) \right\|^2 \right] \quad (15)$$

*with respect to the potential energy* $\Phi$, *is equivalent to solving the weak formulation (14) of the probabilistic Poisson's equation (13). Here,* $\| \cdot \|$ *denotes the Euclidean norm, and* $\operatorname{Cov}$ *denotes the covariance.*

*Furthermore, the variational problem (15) has a unique solution if for all energy functions* $\Phi \in \mathcal{H}_0^1(\Omega; \bar{\rho})$, *the data likelihood homotopy* $\bar{\rho}$ *satisfy the Poincaré inequality:*

$$\mathbb{E}_{\bar{\rho}(x; t)} \left[ \left\| \nabla \Phi^*(x) \right\|^2 \right] \geq \lambda \, \mathbb{E}_{\bar{\rho}(x; t)} \left[ \left\| \Phi^*(x) \right\|^2 \right] \quad (16)$$

*for some positive scalar constant* $\lambda > 0$ *(spectral gap).*

*Proof.* Refer to Appendix B.3. □

As a result, by applying Propositions 2 and 3, we have reformulated the intractable task of minimizing the KLD between flow-driven prior and data likelihood homotopy equivalently as a variational problem with energy loss function (15). By optimizing the potential energy function $\Phi^*$ with respect

to the energy loss and transporting the prior samples through the SDE (10), the prior particles follow a trajectory that accurately approximates the data likelihood homotopy. In doing so, the potential flow $\nabla\Phi^*$ drives the prior samples to posterior regions densely populated with data, thus enabling us to perform generative modelling. In particular, the covariance functional in (15) plays an important role by ensuring that the normalized innovation (residual sum of squares) is inversely proportional to the potential energy. As a result, the potential-generated velocity field $\nabla\Phi^*$ consistently points in the direction of greatest potential ascent, thereby driving the flow of prior particles towards high likelihood regions of the true posterior.

Rather than being an ad hoc addition, the L2 norm $\mathbb{E}_{\bar{\rho}(x;t)}\left[\|\nabla\Phi(x) - \frac{\beta(t)^2}{2}\nabla\log\bar{\rho}(x;t)\|^2\right]$ of (15) arises from the variational formulation. As shown in Vincent (2011), this L2 norm is equivalent to $\mathbb{E}_{\rho(x;\bar{x},t)\,p_{\text{data}}(\bar{x})}\left[\|\nabla\Phi(x) - \frac{\beta(t)^2}{2}\nabla\log\rho(x;\bar{x},t)\|^2\right]$. While it shares similarities with the score matching loss Song & Ermon (2019), the L2 norm here does not serve as the primary loss for training the potential energy $\Phi(x)$. Instead, it serves solely as a regularization term, to stabilize $\Phi(x)$ and account for the diffusion component of the SDE 10. Additionally, it incorporates a weighting of the log-likelihood $\log\bar{\rho}(x;t)$ by $\frac{\beta(t)^2}{2}$, which the hyperparameter $\beta(t)$ is set small to ensure that the diffusion component does not dominate the drift component in the SDE.

Given that the aim is to solve the probabilistic Poisson's equation (13) for all $t$, we include an auxiliary time integral to the energy loss function (15) as follows:

$$\mathcal{L}^{\text{VAPO}}(\theta) = \int_{\mathbb{R}} \mathcal{L}(\theta;t)\,dt = \mathbb{E}_{\mathcal{U}(t;0,1)}\left[\mathcal{L}(\theta;t)\right] \qquad (17)$$

where we have applied Monte Carlo integration, and $\mathcal{U}(a,b)$ denotes the uniform distribution over interval $[a,b]$. In addition, the data likelihood homotopy may not satisfy the Poincaré inequality (16). Hence, we include the right-hand side of the inequality to the loss function (15) to enforce uniqueness of its minimizer. This addtional L2 loss also regularize the energy function, preventing its values from exploding. The spectral gap constant $\lambda$ is left as a training hyperparameter.

In addition, the energy loss (15) requires us to sample from the conditional and data likelihood density homotopies. By design, both the prior $q(x) = \mathcal{N}(x;0,\Lambda)$ and the conditional data likelihood $p(\bar{x}|x) = \mathcal{N}(\bar{x};x,\Pi)$ are assumed to be Gaussian. As a consequence, the Bayes update (4) results in a Gaussian density $\rho(x;\bar{x},t) = \mathcal{N}\big(x;\mu(\bar{x},t),\Sigma(\bar{x},t)\big)$, from which the time-varying mean and covariance can be derived using the Bayes' theorem Bishop (2006), as follows:

$$\mu(\bar{x},t) = t\,\Sigma(t)\,\Pi^{-1}\,\bar{x}, \qquad \Sigma(t) = \left(\Lambda^{-1} + t\,\Pi^{-1}\right)^{-1} \qquad (18)$$

Therefore, to sample from $\rho(x;\bar{x},t)$ or $\rho(x;t)$, we first sample data $x$ from $p_{\text{data}}(\bar{x})$ and compute the mean and covariance according to (18). Then, we can generate samples of the approximate data $x$ using the reparameterization trick $x = \mu(\bar{x},t) + \sqrt{\Sigma}(t)\,\epsilon$, where $\epsilon \sim \mathcal{N}(\epsilon;0,I)$ and $\sqrt{\Sigma}$ is the square root decomposition of $\Sigma$. A detailed derivation of (18) is provided in Appendix B.4.

Nevertheless, parameterizing the conditional homotopy using mean and covariance (18) causes it to converge too quickly to the posterior $\rho(x;\bar{x},1) = p(x|\bar{x})$. As a consequence, most samples are closely clustered around the observed data. To mitigate this issue, a strategy is to slow down its convergence by reparameterizing it with $t + \varepsilon = e^\tau$, where $\tau \in [\ln\varepsilon, \ln(1+\varepsilon)]$. This time reparameterization compels $t + \varepsilon$ to follow a log-uniform (reciprocal) distribution $\mathcal{R}(t+\varepsilon;\varepsilon,1+\varepsilon)$ defined over the interval $[\varepsilon, 1+\varepsilon]$. Here, the hyperparameter $\varepsilon$ is a small positive constant that determines the sharpness of the log-uniform density, and the rate at which its tail decays to zero.

Here, the potential energy $\Phi_\theta$ is parameterized as deep neural networks with parameters $\theta$. Incorporating all of the above considerations, the final energy loss function becomes:

$$\mathcal{L}^{\text{VAPO}}(\theta) = \frac{1}{2}\,\mathbb{E}_{\mathcal{R}(t+\varepsilon;\,\varepsilon,1+\varepsilon)}\left[\mathcal{L}(\theta;t)\right] \qquad (19)$$

where

$$\mathcal{L}(\theta;t) = \text{Cov}_{\rho(x;\bar{x},t)\,p_{\text{data}}(\bar{x})}\left[\Phi_\theta(x),\gamma(x,\bar{x})\right] + \mathbb{E}_{\rho(x;\bar{x},t)\,p_{\text{data}}(\bar{x})}\left[\left\|\nabla\Phi_\theta^*\right\|^2 + \lambda\left\|\Phi_\theta^*\right\|^2\right] \qquad (20)$$

and $\Phi_\theta^* = \Phi_\theta(x) - \frac{\beta(t)^2}{2}\log\rho(x;\bar{x},t)$. Given that the log-likelihood does not depends on model parameters $\theta$, we also substitute $\Phi_\theta^*$ with $\Phi_\theta$ in the covariance functional. The training algorithm of VAPO is given by Algorithm 1 in the Appendix.

Table 1: Comparison of FID scores on unconditional CIFAR-10 image generation. FID baselines are obtained from Zhu et al. (2024).

| EBM-Based Methods | FID ↓ | Other Likelihood-Based Methods | FID ↓ |
|---|---|---|---|
| EBM-SR Nijkamp et al. (2019) | 44.5 | VAE Kingma & Welling (2022) | 78.4 |
| JEM Grathwohl et al. (2020a) | 38.4 | PixelCNN Salimans et al. (2017) | 65.9 |
| EBM-IG Du & Mordatch (2019) | 38.2 | PixelIQN Ostrovski et al. (2018) | 49.5 |
| EBM-FCE Gao et al. (2020) | 37.3 | ResidualFlow Chen et al. (2019a) | 47.4 |
| CoopVAEBM Xie et al. (2021b) | 36.2 | Glow Kingma & Dhariwal (2018) | 46.0 |
| CoopNets Xie et al. (2020) | 33.6 | DC-VAE Parmar et al. (2021) | 17.9 |
| Divergence Triangle Han et al. (2019) | 30.1 | *GAN-Based Methods* | |
| VERA Grathwohl et al. (2021) | 27.5 | WGAN-GP Gulrajani et al. (2017) | 36.4 |
| EBM-CD Du et al. (2021) | 25.1 | SN-GAN Miyato et al. (2018) | 21.7 |
| GEBM Arbel et al. (2021) | 19.3 | SNGAN-DDLS Hill et al. (2022) | 15.4 |
| HAT-EBM Hill et al. (2022) | 19.3 | BigGAN Brock et al. (2019) | 14.8 |
| CF-EBM Zhao et al. (2020) | 16.7 | *Score-Based and Diffusion Methods* | |
| CoopFlow Xie et al. (2022) | 15.8 | NCSN Song & Ermon (2019) | 25.3 |
| CLEL-base Lee et al. (2022) | 15.3 | NCSN-v2 Song & Ermon (2020) | 10.9 |
| VAEBM Xiao et al. (2021) | 12.2 | Action Matching Neklyudov et al. (2023) | 10.0 |
| DRL Gao et al. (2021) | 9.58 | DDPM Distil. Luhman & Luhman (2021) | 9.36 |
| DDAEBM Geng et al. (2024) | 4.82 | Flow Matching Lipman et al. (2023) | 6.35 |
| VAPO (Ours) | **15.4** | Rectified Flow Liu et al. (2023b) | 3.17 |
| VAPO-T (Ours) | **8.33** | NCSN++ Song et al. (2021) | 2.20 |

## 4 EXPERIMENTS

In this section, we show that VAPO is an effective generative model for images. In particular, Section 4.1 demonstrates that VAPO is capable of generating realistic unconditional images on the CIFAR-10 and CelebA datasets. Section 4.2 demonstrates that VAPO is capable of performing smooth interpolation between two generated samples. Apart from that, we also show that VAPO exhibits extensive mode coverage and robustness to anomalous data, and generalizes well to unseen test data. Specifically, Appendix D.1 evaluates model over-fitting and generalization based on the energy histogram of CIFAR-10 train and test sets and the nearest neighbors of generated samples. Appendix D.2 examines robustness to anomalous data by assessing its performance on out-of-distribution (OOD) detection on various image datasets. Appendix E evaluates the convergence of the flow-driven approximate likelihood density and its image samples on long-run sampling. Appendix F accesses the capability of VAPO for compositional generation on the CelebA dataset.

Here, we include two model variants, namely VAPO-A and VAPO-T. Specifically, VAPO-A considers an autonomous (independent of time) energy $\Phi(x(t))$ and VAPO-T considers the more general time-varying energy $\Phi(x(t), t)$. Here, we parameterize VAPO-A using the WideResNet architecture of Xiao et al. (2021) and VAPO-T using the Unet architecture of Du et al. (2023). The performance of VAPO-T are evaluated only on image generation and compositional generation. Implementation details, including model architecture, training, numerical SDE solver, datasets and FID evaluation are provided in Appendix C.

### 4.1 UNCONDITIONAL IMAGE GENERATION

Figure 2 shows the uncurated and unconditional image samples generated from the learned energy model on the CIFAR-10 and CelebA $64 \times 64$ datasets. More generated samples are provided in Appendix G. The samples are of decent quality and resemble the original datasets despite not having the highest fidelity as achieved by state-of-the-art models. Tables 1 and 2 summarize the quantitative evaluations of our proposed framework in terms of FID Heusel et al. (2017) scores on the CIFAR-10 and CelebA datasets. In particular, the VAPO-T model achieved a competitive FID that is better than the majority of existing EBM-based generative models on CIFAR-10. Notably, VAPO-T significantly outperforms VAPO-A, consistent with the results of Salimans & Ho (2021), which show that modifying the EBM architecture from ResNet to Unet results in improved FID performance.

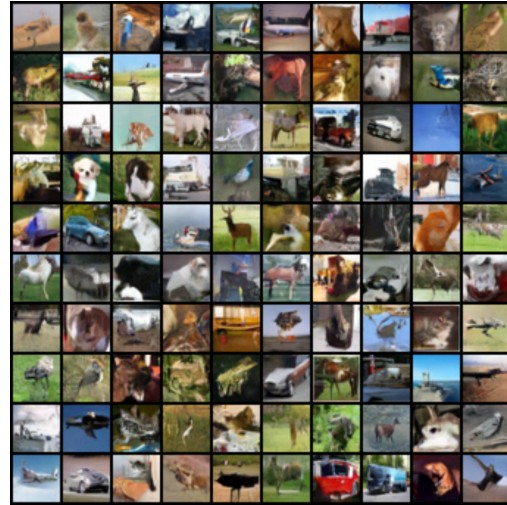 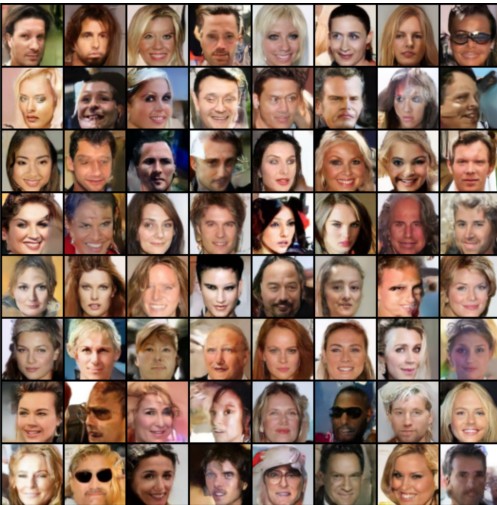

Figure 2: Generated samples on unconditional CIFAR-10 $32 \times 32$ (left) and CelebA $64 \times 64$ (right).

Nevertheless, having dispensed with the implicit MCMC sampling, both VAPO-A and VAPO-T still outperform most of the EBM approaches without relying on complementary latent models or cooperative training. On CelebA, VAPO-A obtains an FID that outperforms some existing EBMs but falls short compared to Song & Ermon (2020) and state-of-the-art models.

### 4.2 IMAGE INTERPOLATION

Figure 3 shows the interpolation results between pairs of generated CelebA samples, where it demonstrates that the VAPO-A model is capable of smooth and semantically meaningful image interpolation. To perform interpolation for two samples $x_1(1)$ and $x_2(1)$, we construct a spherical interpolation between the initial Gaussian noise $x_1(0)$ and $x_2(0)$, and subject them to sampling over the SDE. More interpolation results on CIFAR-10 and CelebA are provided in Appendix G.

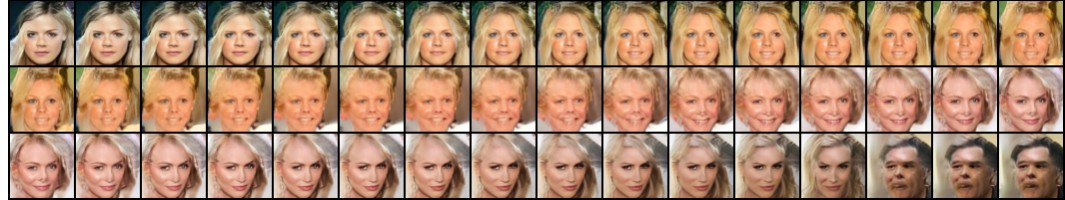

Figure 3: Interpolation results between the leftmost and rightmost generated CelebA $64 \times 64$ samples.

### 5 CONCLUSION

We propose VAPO, a novel energy-based generative modelling framework without the need for expensive and unstable MCMC runs amidst training. Despite the improvement over the majority of existing EBMs, there is still a large performance gap between VAPO and the state-of-the-art score-based (or diffusion) and Poisson flow models Song et al. (2021); Kim et al. (2021); Xu et al. (2022). To close this gap, diffusion recovery likelihood Gao et al. (2021); Zhu et al. (2024), which is shown to be more tractable than marginal likelihood, can be incorporated into the VAPO framework for a more controlled diffusion-guided energy optimization. The dimensionality augmentation technique of Xu et al. (2022; 2023) can also be integrated given that fundamentally, both Poisson flow and VAPO aim to model potential field governed by a Poisson's equation. On top of that, the scalability of VAPO to higher resolution images and its generalizability to other data modalities have yet to be validated. In addition, the current VAPO framework does not allow for class-conditional generation. Moreover, the model training requires a large number of iterations to converge and thus warrants improvement. These important aspects are earmarked for future extensions of our work.

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

## A  RELATED WORKS

Flow-based and diffusion generative models have been pivotal in recent advancements in generative modeling, with a variety of innovative approaches enhancing their effectiveness and applicability. Denoising Diffusion Probabilistic Models (DDPM) Kim et al. (2021) and Score-Based Generative Models (NCSN++) Song et al. (2021) have demonstrated the efficacy of stochastic differential equations (SDEs) in generating high-quality samples through the progressive scaling and refinement of noise. Building upon these foundational techniques, Rectified Flow Liu et al. (2023b) enhances the efficiency of data generation by learning rectified paths to streamlining the diffusion process for rapid generative sampling.

In parallel, Flow Matching Lipman et al. (2023) focuses on learning the evolution of time-dependent diffeomorphic mappings that transport a prior density to a target density, effectively generalizing beyond the probability paths constructed by simple diffusion through the leverage of optimal transport. Stochastic Interpolants Albergo & Vanden-Eijnden (2023) enhance conventional normalizing flows Rezende & Mohamed (2015); Chen et al. (2018) by incorporating stochastic processes that interpolate between prior and target densities, thereby increasing their expressiveness and enabling the handling of more complex data geometries. Schrödinger Bridge Matching Shi et al. (2023) integrates optimal transport theory with diffusion processes through the Schrödinger bridge framework, employing iterative Markovian fitting to construct a probabilistic model that aligns densities, while regularizing entropy for a smoother density transition. Inspired by electrostatics, the Poisson Flow Generative Model (PFGM) Xu et al. (2022) extends flow-based generative models by integrating the Poisson

equation for learning an invertible Poisson flow within an augmented space that includes an additional dimension.

Additionally, Action Matching Neklyudov et al. (2023) introduces a novel paradigm by learning the dynamics of the generative process through the optimization of the potential action (energy) that governs the underlying gradient field. This perspective enables a more physically grounded approach to modeling stochastic dynamics within generative frameworks. On the other hand, approaches such as Diffusion Recovery Likelihood (DRL) Gao et al. (2021) and Denoising Diffusion Adversarial Energy-Based Models (DDAEBM) Geng et al. (2024) harness diffusion processes to enhance the sampling efficiency and training stability of conventional energy-based models (EBMs). Together, these models represent significant strides in integrating diffusion, flow-based, optimal transport, and energy-based methodologies for generative modeling.

Nevertheless, existing generative models have not incorporated log-homotopy transformations or a variational (Deep Ritz) formulation that aligns the Fokker–Planck (or continuity) equation directly with the evolution of a time-dependent density homotopy. Furthermore, our proposed VAPO loss is formulated in terms of potential energy, with energy learning primarily governed by the bilinear covariance operator. This formulation offers direct control over the energy training, a characteristic inherent to the contrastive divergence loss Hinton (2002) of EBMs but absent in score matching Vincent (2011). Such control enables potential applications in safe image generation Schramowski et al. (2023), eliminating the need for post hoc guidance techniques.

## B  PROOFS AND DERIVATIONS

### B.1  PROOF OF PROPOSITION 1

*Proof.* Differentiating the conditional homotopy $\rho(x; \bar{x}, t)$ in (4) with respect to $t$, we have

$$
\begin{aligned}
\frac{\partial \rho(x; \bar{x}, t)}{\partial t} &= \frac{1}{\int_\Omega e^{f(x; \bar{x}, t)} \, dx} \frac{\partial [e^{f(x; \bar{x}, t)}]}{\partial t} - \frac{e^{f(x; \bar{x}, t)}}{[\int_\Omega e^{f(x; \bar{x}, t)} \, dx]^2} \frac{\partial [\int_\Omega e^{f(x; \bar{x}, t)} \, dx]}{\partial t} \\
&= \frac{1}{\int_\Omega e^{f(x; \bar{x}, t)} \, dx} \frac{\partial [e^{f(x; \bar{x}, t)}]}{\partial f} \frac{\partial f(x; \bar{x}, t)}{\partial t} - \frac{e^{f(x; \bar{x}, t)}}{[\int_\Omega e^{f(x; \bar{x}, t)} \, dx]^2} \int_\Omega \frac{\partial [e^{f(x; \bar{x}, t)}]}{\partial f} \frac{\partial f(x; \bar{x}, t)}{\partial t} \, dx \\
&= \frac{e^{f(x; \bar{x}, t)}}{\int_\Omega e^{f(x; \bar{x}, t)} \, dx} \frac{\partial f(x; \bar{x}, t)}{\partial t} - \frac{e^{f(x; \bar{x}, t)}}{\int_\Omega e^{f(x; \bar{x}, t)} \, dx} \int_\Omega \frac{e^{f(x; \bar{x}, t)}}{\int_\Omega e^{f(x; \bar{x}, t)} \, dx} \frac{\partial f(x; \bar{x}, t)}{\partial t} \, dx \\
&= \rho(x; \bar{x}, t) \left( \frac{\partial f(x; \bar{x}, t)}{\partial t} - \int_\Omega \rho(x; \bar{x}, t) \frac{\partial f(x; \bar{x}, t)}{\partial t} \, dx \right) \\
&= -\frac{1}{2} \rho(x; \bar{x}, t) \left( (x - \bar{x})^T \, \Pi^{-1} \, (x - \bar{x}) - \int_\Omega \rho(x; \bar{x}, t) \, (x - \bar{x})^T \, \Pi^{-1} \, (x - \bar{x}) \, dx \right)
\end{aligned}
$$

(21)

where we used the quotient rule in the first equation, and chain rule in the second equation.

Let $\gamma(x, \bar{x}) = (x - \bar{x})^T \, \Pi^{-1} \, (x - \bar{x})$ and using the fact that

$$
\frac{\partial \bar{\rho}(x; t)}{\partial t} = \frac{\partial \int_\Omega \rho(x; \bar{x}, t) \, p_{\text{data}}(\bar{x}) \, d\bar{x}}{\partial t} = \int_\Omega \frac{\partial \rho(x; \bar{x}, t)}{\partial t} \, p_{\text{data}}(\bar{x}) \, d\bar{x}
$$

(22)

we can substitute (21) into (22) to get

$$
\frac{\partial \bar{\rho}(x; t)}{\partial t} = -\frac{1}{2} \int_\Omega p_{\text{data}}(\bar{x}) \, \rho(x; \bar{x}, t) \left( \gamma(x, \bar{x}) - \int_\Omega \rho(x; \bar{x}, t) \, \gamma(x, \bar{x}) \, dx \right) d\bar{x}
$$

(23)

Given that both $\rho(x; \bar{x}, t)$ and $p_{\text{data}}(x)$ are normalized (proper) density functions, writing (23) in terms of expectations yields the PDE in (8). □

### B.2  PROOF OF PROPOSITION 2

Here, we used the Einstein tensor notation interchangeably with the conventional notation for vector dot product and matrix-vector multiplication in PDE.

Given that the context is clear, we write $g_t(\cdot)$ in place of time-varying functions $g(\cdot, t)$. For brevity, we will also omit the time index $t$, and write $g$ in place of $g_t(x)$.

*Proof.* Applying the forward Euler method to the particle flow SDE (10) using step size $\Delta_t$, we obtain:

$$x_{t+\Delta_t} = \alpha_t(x_t) = x_t + \Delta_t\, u(x_t) \tag{24}$$

where

$$u(x_t) = \nabla \Phi^*(x_t) \tag{25}$$

where we denote $x_t$ as the discretizations random variables $x(t)$.

Assuming that the $\alpha_t : \Omega \to \Omega$ is a diffeomorphism (bijective function with differentiable inverse), the push-forward operator $\alpha_{t\#} : \mathbb{R} \to \mathbb{R}$ on density function $\rho_t^\Phi \mapsto \rho_{t+\Delta_t}^\Phi := \alpha_{t\#}\rho_t^\Phi$ is defined by:

$$\int_\Omega \rho_{t+\Delta_t}^\Phi(x)\, g(x)\, dx = \int_\Omega \alpha_{t\#}\rho_t^\Phi(x)\, g(x)\, dx = \int_\Omega \rho_t^\Phi(x)\, g\big(\alpha_t(x)\big)\, dx \tag{26}$$

for any measurable function $g$.

Associated with the change-of-variables formula (26) is the following density transformation:

$$\rho_{t+\Delta_t}^\Phi\big(\alpha_t(x)\big) = \frac{1}{|D\alpha_t|}\, \rho_t^\Phi(x) \tag{27}$$

where $|D\alpha_t|$ denotes the Jacobian determinant of $\alpha_t$.

From (8) and (23), we have

$$\frac{\partial \ln \bar{\rho}_t(x)}{\partial t} = \frac{1}{\bar{\rho}_t(x)}\frac{\partial \bar{\rho}_t(x)}{\partial t} = -\frac{1}{\bar{\rho}_t(x)}\frac{1}{2}\, \mathbb{E}_{p_{\text{data}}(\bar{x})}\Big[\rho_t(x,\bar{x})\big(\gamma(x,\bar{x}) - \bar{\gamma}(x,\bar{x})\big)\Big] \tag{28}$$

Applying the forward Euler method to (28), we obtain

$$\ln \bar{\rho}_{t+\Delta_t}(x) \geq \ln \bar{\rho}_t(x) - \frac{\Delta_t}{2}\frac{1}{\bar{\rho}_t(x)}\, \mathbb{E}_{p_{\text{data}}(\bar{x})}\Big[\rho_t(x,\bar{x})\big(\gamma(x,\bar{x}) - \bar{\gamma}(x,\bar{x})\big)\Big] \tag{29}$$

Applying the change-of-variables formula (26) and density transformation (27), then substituting (29) into the KLD $\mathcal{D}_{\text{KL}}\big[\rho_{t+\Delta_t}^\Phi \,\|\, \bar{\rho}_{t+\Delta_t}\big]$ at time $t + \Delta_t$, we have

$$\mathcal{D}_{\text{KL}}\big[\rho_{t+\Delta_t}^\Phi(x) \,\|\, \bar{\rho}_{t+\Delta_t}(x)\big] = \int_\Omega \rho_t^\Phi(x)\ln\left(\frac{\rho_{t+\Delta_t}^\Phi\big(\alpha_t(x)\big)}{\bar{\rho}_{t+\Delta_t}\big(\alpha_t(x)\big)}\right) dx$$

$$= \int_\Omega \rho_t^\Phi(x)\left(\ln \rho_t^\Phi(x) - \ln|D\alpha_t| - \ln \bar{\rho}_t\big(\alpha_t(x)\big)\right.$$

$$\left. + \frac{\Delta_t}{2}\frac{1}{\bar{\rho}_t\big(\alpha_t(x)\big)}\, \mathbb{E}_{p_{\text{data}}(\bar{x})}\Big[\rho_t\big(\alpha_t(x),\bar{x}\big)\big(\gamma\big(\alpha_t(x),\bar{x}\big) - \bar{\gamma}\big(\alpha_t(x),\bar{x}\big)\big)\Big] + C\right) dx \tag{30}$$

Consider minimizing the KLD (30) with respect to $\alpha_t$ as follows:

$$\min_{\alpha_t} \mathcal{D}_{\text{KL}}(\alpha_t)$$

$$= \min_{\alpha_t} \underbrace{\frac{\Delta_t}{2}\int_\Omega \rho_t^\Phi(x)\frac{1}{\bar{\rho}_t\big(\alpha_t(x)\big)}\, \mathbb{E}_{p_{\text{data}}(\bar{x})}\Big[\rho_t\big(\alpha_t(x),\bar{x}\big)\big(\gamma\big(\alpha_t(x),\bar{x}\big) - \bar{\gamma}\big(\alpha_t(x),\bar{x}\big)\big)\Big] dx}_{\mathcal{D}_1^{\text{KL}}(\alpha_t)}$$

$$\underbrace{- \int_\Omega \rho_t^\Phi(x)\ln \bar{\rho}_t\big(\alpha_t(x)\big)\, dx}_{\mathcal{D}_2^{\text{KL}}(\alpha_t)} \underbrace{- \int_\Omega \rho_t^\Phi(x)\ln|D\alpha_t|\, dx}_{\mathcal{D}_3^{\text{KL}}(\alpha_t)} \tag{31}$$

where we have neglected the constant terms that do not depend on $\alpha_t$.

To solve the optimization (31), we consider the following optimality condition in the first variation of $\mathcal{D}_{\mathrm{KL}}$:

$$\mathcal{I}(\alpha, \nu) = \frac{d}{d\epsilon} \, \mathcal{D}_{\mathrm{KL}} \left( \alpha(x) + \epsilon \, \nu(x) \right) \bigg|_{\epsilon=0} = 0 \tag{32}$$

which must hold for all trial function $\nu(x)$.

Taking the variational derivative of the first functional $\mathcal{D}_1^{\mathrm{KL}}$ in (31), we have

$$\mathcal{I}^1(\alpha, \nu) = \frac{d}{d\epsilon} \, \mathcal{D}_1^{\mathrm{KL}}(\alpha + \epsilon\nu) \bigg|_{\epsilon=0}$$

$$= \frac{\Delta}{2} \int_\Omega \rho^\Phi(x) \frac{d}{d\epsilon} \left\{ \frac{1}{\bar{\rho}(\alpha + \epsilon\nu)} \, \mathbb{E}_{p_{\text{data}}(\bar{x})} \left[ \rho(\alpha + \epsilon\nu, \bar{x}) \left( \gamma(\alpha + \epsilon\nu, \bar{x}) - \bar{\gamma}(\alpha + \epsilon\nu, \bar{x}) \right) \right] \right\} \bigg|_{\epsilon=0} dx$$

$$= \frac{\Delta}{2} \int_\Omega \rho^\Phi(x) \, D \left\{ \frac{1}{\bar{\rho}(x)} \, \mathbb{E}_{p_{\text{data}}(\bar{x})} \left[ \rho(x, \bar{x}) \left( \gamma(x, \bar{x}) - \bar{\gamma}(x, \bar{x}) \right) \right] \right\} \nu \, dx$$

$$\tag{33}$$

where $Dg := \nabla^T g$ denotes the Jacobian of function $g(x)$ with respect to $x$.

A Taylor series expansion of the derivative $\frac{\partial g}{\partial x_i}(\alpha)$ with respect to $x_i$ yields

$$\frac{\partial g(\alpha)}{\partial x_i} = \frac{\partial g(x + \Delta u)}{\partial x_i} = \frac{\partial g(x)}{\partial x_i} + \Delta \sum_j \frac{\partial^2 g(x)}{\partial x_i \, \partial x_j} \, u_j + O(\Delta^2) \tag{34}$$

Using the Taylor series expansion (34), (33) can be written in tensor notation as follows:

$$\mathcal{I}^1(\alpha, \nu) = \frac{\Delta}{2} \int_\Omega \rho^\Phi(x) \sum_i \frac{\partial}{\partial x_i} \left\{ \frac{1}{\bar{\rho}(x)} \, \mathbb{E}_{p_{\text{data}}(\bar{x})} \left[ \rho(x; \bar{x}) \left( \gamma(x, \bar{x}) - \bar{\gamma}(x, \bar{x}) \right) \right] \right\} \nu_i \, dx \; + \; O(\Delta^2)$$

$$\tag{35}$$

Taking the variational derivative of the second functional $\mathcal{D}_2^{\mathrm{KL}}$ in (31) yields

$$\mathcal{I}^2(\alpha, \nu) = \frac{d}{d\epsilon} \, \mathcal{D}_2^{\mathrm{KL}}(\alpha + \epsilon\nu) \bigg|_{\epsilon=0}$$

$$= \int_\Omega \rho^\Phi(x) \frac{d}{d\epsilon} \ln \bar{\rho}(\alpha + \epsilon\nu) \bigg|_{\epsilon=0} dx$$

$$= \int_\Omega \rho^\Phi(x) \frac{1}{\bar{\rho}(\alpha)} \nabla \bar{\rho}(\alpha) \cdot \nu \, dx \tag{36}$$

$$= \int_\Omega \rho^\Phi(x) \nabla \ln \bar{\rho}(\alpha) \cdot \nu \, dx$$

where we have used the derivative identity $d \ln g = \frac{1}{g} \, dg$ to obtain the second equation.

Using the Taylor series expansion (34), (36) can be written in tensor notation as follows:

$$\mathcal{I}^2(\alpha, \nu) = -\int_\Omega \rho^\Phi(x) \sum_i \left( \frac{\partial \ln \bar{\rho}(x)}{\partial x_i} - \Delta \sum_j \frac{\partial^2 \ln \bar{\rho}(x)}{\partial x_i \, \partial x_j} \, u_j \right) \nu_i \, dx \; + \; O(\Delta^2)$$

$$\tag{37}$$

$$= -\int_\Omega \rho^\Phi(x) \sum_i \left( \frac{\partial \ln \bar{\rho}(x)}{\partial x_i} - \Delta \sum_j \frac{\partial^2 \ln \bar{\rho}(x)}{\partial x_i \, \partial x_j} \, u_j \right) \nu_i \, dx \; + \; O(\Delta^2)$$

Similarly, taking the variational derivative of the $\mathcal{D}_3^{\mathrm{KL}}$ term in (31), we have

$$
\begin{aligned}
\mathcal{I}^3(\alpha, \nu) &= \frac{d}{d\epsilon} \mathcal{D}_3^{\mathrm{KL}}(\alpha + \epsilon\nu) \Big|_{\epsilon=0} \\
&= \int_\Omega \rho^\Phi(x) \frac{d}{d\epsilon} \ln |D(\alpha + \epsilon\nu)| \Big|_{\epsilon=0} dx \\
&= \int_\Omega \rho^\Phi(x) \frac{1}{|D\alpha|} \frac{d}{d\epsilon} |D(\alpha + \epsilon\nu)| \Big|_{\epsilon=0} dx \\
&= \int_\Omega \rho^\Phi(x) \operatorname{tr} \left( D\alpha^{-1} D\nu \right) dx
\end{aligned}
\tag{38}
$$

where we have used the following Jacobi's formula:

$$
\frac{d}{d\epsilon} |D(\alpha + \epsilon\nu)| \Big|_{\epsilon=0} = |D\alpha| \operatorname{tr} \left( D\alpha^{-1} D\nu \right)
\tag{39}
$$

to obtain the last equation in (38).

The inverse of Jacobian $D\alpha^{-1}$ can be expanded via Neuman series to obtain

$$
D\alpha^{-1} = \left( \mathrm{I} + \Delta \, Du \right)^{-1} = \mathrm{I} - \Delta \, Du \, + \, O(\Delta^2)
\tag{40}
$$

Substituting in (40) and using the Taylor series expansion (34), (36) can be written in tensor notation as follows:

$$
\begin{aligned}
\mathcal{I}^3(\alpha, \nu) &= \int_\Omega \sum_i \left( \rho^\Phi(x) \frac{\partial \nu_i}{\partial x_i} - \Delta \sum_j \rho^\Phi(x) \frac{\partial u_j}{\partial x_i} \frac{\partial \nu_i}{\partial x_j} \right) dx \, + \, O(\Delta^2) \\
&= \int_\Omega \sum_i \left( \frac{\partial \rho^\Phi(x)}{\partial x_i} \nu_i - \Delta \sum_j \frac{\partial}{\partial x_j} \left\{ \rho^\Phi(x) \frac{\partial u_j}{\partial x_i} \right\} \nu_i \right) dx \, + \, O(\Delta^2) \\
&= \int_\Omega \sum_i \left( \frac{\partial \rho^\Phi(x)}{\partial x_i} - \Delta \sum_j \frac{\partial}{\partial x_j} \left\{ \rho^\Phi(x) \frac{\partial u_j}{\partial x_i} \right\} \right) \nu_i \, dx \, + \, O(\Delta^2)
\end{aligned}
\tag{41}
$$

where we have used integration by parts to obtain the second equation.

Taking the limit $\lim \Delta \to 0$, the terms $O(\Delta^2)$ that approach zero exponentially vanish. Subtracting (35) by (37) and (41) then equating to zero, we obtain the first-order optimality condition (32) as follows:

$$
\begin{aligned}
\int_\Omega \bar{\rho}(x) \sum_i \Bigg( \sum_j - \frac{\partial}{\partial x_i} \left\{ \frac{1}{\bar{\rho}(x)} \frac{\partial}{\partial x_j} \left\{ \bar{\rho}(x) \, u_j \right\} \right\} \\
+ \frac{1}{2} \frac{\partial}{\partial x_i} \left\{ \frac{1}{\bar{\rho}(x)} \mathbb{E}_{p_{\mathrm{data}}(\bar{x})} \left[ \rho(x; \bar{x}) \left( \gamma(x, \bar{x}) - \bar{\gamma}(x, \bar{x}) \right) \right] \right\} \Bigg) \nu_i \, dx = 0
\end{aligned}
\tag{42}
$$

where we have assumed that $\rho^\Phi(x) \equiv \bar{\rho}(x)$ holds, and used the following identities:

$$
\begin{aligned}
\frac{\partial \ln \bar{\rho}(x)}{\partial x_i} &= \frac{1}{\bar{\rho}(x)} \frac{\partial \bar{\rho}(x)}{\partial x_i} \\
\frac{\partial^2 \ln \bar{\rho}(x)}{\partial x_i \, \partial x_j} &= \frac{\partial}{\partial x_i} \left( \frac{1}{\bar{\rho}(x)} \frac{\partial \bar{\rho}(x)}{\partial x_j} \right)
\end{aligned}
\tag{43}
$$

Given that $\nu_i$ can take any value, the equation (42) holds (in the weak sense) only if the terms within the round bracket vanish. Integrating this term with respect to the $x_i$, we are left with

$$
\sum_j \frac{\partial}{\partial x_j} \left\{ \bar{\rho}(x) \, u_j \right\} = \frac{1}{2} \mathbb{E}_{p_{\mathrm{data}}(\bar{x})} \left[ \rho(x; \bar{x}) \left( \gamma(x, \bar{x}) - \bar{\gamma}(x, \bar{x}) \right) \right] \, + \, \bar{\rho}(x) \, C
\tag{44}
$$

which can also be written in vector notation as follows:

$$\nabla \cdot \big(\bar{\rho}(x)\, u\big) = \frac{1}{2}\, \mathbb{E}_{p_{\text{data}}(\bar{x})} \Big[\rho(x;\bar{x})\, \big(\gamma(x,\bar{x}) - \bar{\gamma}(x,\bar{x})\big)\Big] \,+\, \bar{\rho}(x)\, C \tag{45}$$

To find the scalar constant $C$, we integrate both sides of (45) to get

$$
\begin{aligned}
\int_{\Omega} \nabla \cdot \big(\bar{\rho}(x)\, u\big)\, dx &= \frac{1}{2} \int_{\Omega} \mathbb{E}_{p_{\text{data}}(\bar{x})} \Big[\rho(x;\bar{x})\, \big(\gamma(x,\bar{x}) - \bar{\gamma}(x,\bar{x})\big)\Big]\, dx \,+\, \int_{\Omega} \bar{\rho}(x)\, C\, dx \\
&= \frac{1}{2} \int_{\Omega} \mathbb{E}_{p_{\text{data}}(\bar{x})} \Big[\rho(x;\bar{x})\, \big(\gamma(x,\bar{x}) - \bar{\gamma}(x,\bar{x})\big)\Big]\, dx \,+\, C
\end{aligned}
\tag{46}
$$

Applying the divergence theorem to the left-hand side of (46), we have

$$\int_{\Omega} \nabla \cdot \big(\bar{\rho}(x)\, u\big)\, dx = \int_{\partial\Omega} \bar{\rho}(x)\, u \cdot \hat{n}\, dx \tag{47}$$

where $\hat{n}$ is the outward unit normal vector to the boundary $\partial\Omega$ of $\Omega$.

Given that $\bar{\rho}(x)$ is a normalized (proper) density with compact support (vanishes on the boundary), the term (47) becomes zero and we obtain $C = 0$. Substituting this and $u(x) = \nabla\Phi^*(x)$ into (45), we arrive at the PDE

$$\nabla \cdot \big(\bar{\rho}_t(x)\, \nabla\Phi^*(x)\big) = \frac{1}{2}\, \mathbb{E}_{p_{\text{data}}(\bar{x})} \Big[\rho_t(x,\bar{x})\, \big(\gamma(x,\bar{x}) - \bar{\gamma}(x,\bar{x})\big)\Big] \tag{48}$$

Assume that the base case $\rho_0^{\Phi}(x) \equiv \bar{\rho}_0(x)$ holds, and that there exists a solution to (48) for every $t$. The proposition follows by the principle of induction.

$\square$

### B.3 PROOF OF PROPOSITION 3

*Proof.* The energy loss function in (15) can be written as follows:

$$\mathcal{L}(\Phi^*, t) = \frac{1}{2}\, \mathbb{E}_{\rho(x;\bar{x},t)\, p_{\text{data}}(x)} \Big[\Phi^*(x)\, \big(\gamma(x,\bar{x}) - \bar{\gamma}(x,\bar{x})\big)\Big] + \frac{1}{2}\, \mathbb{E}_{\bar{\rho}(x;t)} \Big[\big\|\nabla\Phi^*(x)\big\|^2\Big] \tag{49}$$

where we have assumed, without loss of generality, that a normalized potential energy $\bar{E}_\theta(x;t) = 0$. For an unnormalized solution $\Phi^*(x)$, we can always obtain the desired normalization by subtracting its mean.

The optimal solution $\Phi^*$ of the functional (49) is given by the first-order optimality condition:

$$\mathcal{I}(\Phi^*, \Psi) = \frac{d}{d\epsilon}\, \mathcal{L}(\Phi^*(x) + \epsilon\Psi(x), t)\Big|_{\epsilon=0} = 0 \tag{50}$$

which must hold for all trial function $\Psi$.

Taking the variational derivative of the particle flow objective (50) with respect to $\epsilon$, we have

$$
\begin{aligned}
\mathcal{I}(\Phi^*, \Psi) &= \frac{d}{d\epsilon}\, \mathcal{L}(\Phi^* + \epsilon\Psi)\Big|_{\epsilon=0} \\
&= \frac{1}{2} \int_{\Omega\times\Omega} p_{\text{data}}(x)\, \rho(x;\bar{x})\, \big(\gamma(x,\bar{x}) - \bar{\gamma}(x,\bar{x})\big)\, \frac{d}{d\epsilon}(\Phi^* + \epsilon\Psi)\, d\bar{x}\, dx \\
&\quad + \frac{1}{2} \int_{\Omega} \bar{\rho}(x)\, \frac{d}{d\epsilon}\big\|\nabla(\Phi^* + \epsilon\Psi)\big\|^2\, dx \\
&= \frac{1}{2} \int_{\Omega\times\Omega} p_{\text{data}}(x)\, \rho(x;\bar{x})\, \big(\gamma(x,\bar{x}) - \bar{\gamma}(x,\bar{x})\big)\, \Psi\, d\bar{x}\, dx \,+\, \int_{\Omega} \bar{\rho}(x)\, \nabla\Phi^* \cdot \nabla\Psi\, dx
\end{aligned}
\tag{51}
$$

Given that $\Phi^* \in \mathcal{H}_0^1(\Omega; \bar{\rho})$, its value vanishes on the boundary $\partial\Omega$. Therefore, the second summand of the last expression in (51) can be written, via multivariate integration by parts, as

$$\int_{\Omega} \bar{\rho}(x)\, \nabla\Phi^* \cdot \nabla\Psi = -\int_{\Omega} \nabla \cdot \big(\bar{\rho}(x)\, \nabla\Phi^*\big)\, \Psi\, dx \tag{52}$$

By substituting (52) into (51), we get

$$\mathcal{I}(\Phi^*, \Psi) = \frac{1}{2} \int_\Omega \int_\Omega p_{\text{data}}(x)\rho(x;\bar{x}) \left(\gamma(x,\bar{x}) - \bar{\gamma}(x,\bar{x})\right) \Psi \, d\bar{x} \, dx \; - \; \int_\Omega \nabla \cdot \left(\bar{\rho}(x) \nabla \Phi^*\right) \Psi \, dx$$

$$= \int_\Omega \left( \frac{1}{2} \int_\Omega p_{\text{data}}(x)\rho(x;\bar{x}) \left(\gamma(x,\bar{x}) - \bar{\gamma}(x,\bar{x})\right) d\bar{x} \; - \; \int_\Omega \nabla \cdot \left(\bar{\rho}(x) \nabla \Phi^*\right) \right) \Psi \, dx \tag{53}$$

and equating it to zero, we obtain the weak formulation (14) of the probabilistic Poisson's equation.

Given that the Poincaré inequality (16) holds, (Laugesen et al., 2015, Theorem 2.2) presents a rigorous proof of existence and uniqueness for the solution of the weak formulation (14), based on the Hilbert-space form of the Riesz representation theorem. □

### B.4 Derivation of Time-varying Mean and Variance in (18)

Given the following marginal Gaussian distribution for $z$ and a conditional Gaussian distribution for $x$ given $x$, as defined in Section 3.1:

$$q(x) = \mathcal{N}(x; 0, \Lambda) \tag{54a}$$
$$p(\bar{x}|x) = \mathcal{N}(\bar{x}; x, \Pi) \tag{54b}$$

The posterior distribution of $x$ given $\bar{x}$ is obtained via Bayes' theorem as

$$p(x|\bar{x}) = \frac{p(\bar{x}|x)\,q(x)}{\int_\Omega p(\bar{x}|x)\,q(x)\,dx} = \mathcal{N}(x; \mu, \Sigma) \tag{55}$$

and remains a Gaussian, whose mean and variance are given by:

$$\mu(\bar{x}) = \Sigma\,\Pi^{-1}\,\bar{x} \tag{56a}$$
$$\Sigma = \left(\Lambda^{-1} + \Pi^{-1}\right)^{-1} \tag{56b}$$

In fact, the conditional homotopy (4) can be written as

$$\rho(x; \bar{x}, t) = \frac{p(\bar{x}; t|x)\,q(x)}{\int_\Omega p(\bar{x}; t|x)\,q(x)\,dx} \tag{57}$$

where

$$p(\bar{x}; t|x) = \mathcal{N}(x; \mu, \frac{1}{t}\Pi) \tag{58}$$

Notice that the terms involving $t$ in the numerator and denominator of (57) cancel each other out. Substituting the variance of (58) into (57) and using (55)-(56), we obtain (18).

## C  Experimental Details

### C.1  Model architecture

Our network architectures for the model variants VAPO-A and VAPO-T are based on the WideResNet Zagoruyko & Komodakis (2016) and the Unet Ronneberger et al. (2015), respectively. We adopt the model hyperparameters used in Xiao et al. (2021) for WideResNet and Du et al. (2023) for Unet. In particular, we include a spectral regularization loss for model training to penalizes the spectral norm of the convolutional layer in WideResNet. Also, we remove the final scale-by-sigma operation Kim et al. (2021); Song et al. (2021) and replace it with a L2 norm $\frac{1}{2}\|x(t) - f_\theta(x(t))\|^2$ between the input $x(t)$ and the output of the Unet $f_\theta(x(t))$. Here, we replace LeakyReLU activations with Gaussian Error Linear Unit (GELU) activations Hendrycks & Gimpel (2017) for both WideResNet and Unet, which we found improves training stability and convergence. Additionally, we apply weight normalization with data-dependent initialization Salimans & Kingma (2016) on the convolutional layers of WideResNet to regularize the output energy.

---

**Algorithm 1** VAPO Training

---

**Input:** Initial energy model $\Phi_\theta$, spectral gap constant $\lambda$, sharpness constant $\varepsilon$, standard deviation $\omega$ of prior density, standard deviation $\sigma$ of conditional likelihood, $\beta$ diffusion coefficient, and batch size $B$.

**repeat**

    Sample observed data $\bar{x}_i \sim p_{\text{data}}(\bar{x})$, $t_i \sim \mathcal{R}(t + \varepsilon;\ \varepsilon, 1 + \varepsilon)$, and $\epsilon_i \sim \mathcal{N}(\epsilon; 0, I)$

    Sample $x_i \sim \rho(x; \bar{x}, t)$ via reparameterization $x_i = \mu(\bar{x}_i, t_i) + \sqrt{\Sigma}(t_i)\,\epsilon_i$ based on (18)

    Compute gradient $\nabla\Phi_\theta(x_i)$ w.r.t. $x_i$ via backpropagation

    Calculate innovation $\gamma(x_i, \bar{x}_i)$ based on (9)

    Calculate VAPO loss $\mathcal{L}^{\text{VAPO}}(\theta) = \frac{1}{B}\sum_{i=1}^{B}\mathcal{L}(\theta; t_i)$ based on (19)-(20)

    Update energy model parameters $\theta$ with the gradient of $\mathcal{L}^{\text{VAPO}}(\theta)$

**until** $\theta$ converged

---

## C.2 TRAINING

We use the Lamb optimizer You et al. (2020) and a learning rate of 0.001 for all the experiments. We find that Lamb performs better than Adam over large learning rates. We use a batch size of 64 and 32 for training CIFAR-10 and CelebA, respectively. We set the diffusion coefficient to a small constant value $\beta = \sqrt{2} \times 0.005$ to ensure that the drift component is not overwhelmed by the diffusion component in the SDE 10. For all experiments, we set a spectral gap constant $\lambda$ of 0.001, and a sharpness constant $\varepsilon$ of 0.0001 in our training. Here, we set the standard deviation $\omega$ of the prior density to be 1 so that the data likelihood homotopy is variance-preserving. Also, we set the standard deviation $\sigma$ of conditional data likelihood to be 0.01 so that the difference between samples $x$ and data $\bar{x}$ is indistinguishable to human eyes Song & Ermon (2019). All models are trained on a single NVIDIA A100 (80GB) GPU until the FID scores, computed on 2,500 samples, no longer show improvement. We observe that the models converge within 800k training iterations.

## C.3 NUMERICAL SOLVER

In our experiments, we apply the Euler–Maruyama method for numerical solution of the SDE 10 using a step size of 0.01. Given that the energy of VAPO-A is time independent, we are allowed to set a longer SDE time interval, allowing the additional SDE iterations to further refine the samples within regions of high likelihood and improve the quality of generated images. We observe that setting a terminal time of 1.625 for the SDE solver gives the best results for VAPO-A. Nevertheless, this is not feasible to VAPO-T since the time embedding model of Unet is trained explicitly on the predetermined SDE time interval $[0, 1]$.

## C.4 DATASETS

We use the CIFAR-10 Krizhevsky (2009) and CelebA Liu et al. (2015) datasets for our experiments CIFAR-10 is of resolution $32 \times 32$, and contains $50,000$ training images and $10,000$ test images. CelebA contains $202,599$ face images, of which $162,770$ are training images and $19,962$ are test

Table 2: Comparison of FID scores on unconditional CelebA $64 \times 64$. FID baselines obtained from Gao et al. (2021).

| Methods | FID $\downarrow$ |
| --- | --- |
| NCSN Song & Ermon (2019) | 25.3 |
| NCSN-v2 Song & Ermon (2020) | 10.2 |
| EBM-Triangle Han et al. (2020) | 24.7 |
| EBM-SR Nijkamp et al. (2019) | 23.0 |
| Divergence Triangle Han et al. (2019) | 18.2 |
| CoopNets Xie et al. (2020) | 16.7 |
| DDAEBM Geng et al. (2024) | 10.3 |
| VAPO-A (Ours) | **13.4** |

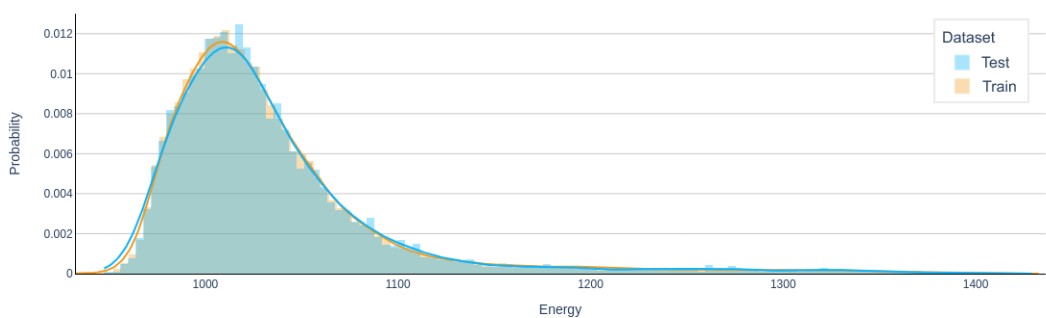

Figure 4: Histogram of energy output for CIFAR-10 train and test set.

images. For processing, we first clip each image to $178 \times 178$ and then resize it to $64 \times 64$. For processing, we first crop each image to a square image whose side is of length which is the minimum of the height and weight, and then we resize it to $64 \times 64$ or $128 \times 128$. For resizing, we set the anti-alias to True. We apply horizontal random flip as data augmentation for the training datasets.

## C.5 QUANTITATIVE EVALUATION

We employ the FID and inception scores as quantitative evaluation metrics for assessing the quality of generated samples. For CIFAR-10, we compute the Frechet distance between $50,000$ samples and the pre-computed statistics on the training set [13]. For CelebA $64 \times 64$, we follow the setting in Song & Ermon (2020) where the distance is computed between $5,000$ samples and the pre-computed statistics on the test set. For model selection, we follow Song et al. (2021) and pick the checkpoint with the smallest FID scores, computed on 2,500 samples every 10,000 iterations.

## D  MODE EVALUATION

In this section, we evaluate the mode coverage and over-fitting of the proposed VAPO framework.

### D.1 MODEL OVER-FITTING AND GENERALIZATION

To assess over-fitting, Figure 4 plots the histogram of the energy outputs on the CIFAR-10 train and test dataset. The energy histogram shows that the learned energy model assigns similar energy values to both train and test set images. This indicates that the VAPO-A model generalizes well to unseen test data and extensively covers all the modes in the training data.

In addition, Figure 5 presents the nearest neighbors of the generated samples in the train set of CIFAR-10. It shows that nearest neighbors are significantly different from the generated samples, thus suggesting that our models do not over-fit the training data and generalize well across the underlying data distribution.

### D.2 OUT-OF-DISTRIBUTION DETECTION

We evaluate robustness of the proposed VAPO framework to anomalous data by assessing its performance on unsupervised out-of-distribution (OOD) detection. Given that potential energy characterizes a stationary Boltzmann distribution, the energy model can be used to distinguish between the in-distribution and out-distribution samples based on the energy values it assigns. In particular, the energy model trained on CIFAR-10 train set is used for assigning normalized energy values to in-distribution samples (CIFAR-10 test set) and out-distribution samples from various other image datasets. The area under the receiver operating characteristic curve (AUROC) is used as a quantitative metric to determine the efficacy of the VAPO-A model in OOD detection, where a high AUROC score indicates that the model correctly assigns low energy to out-distribution samples.

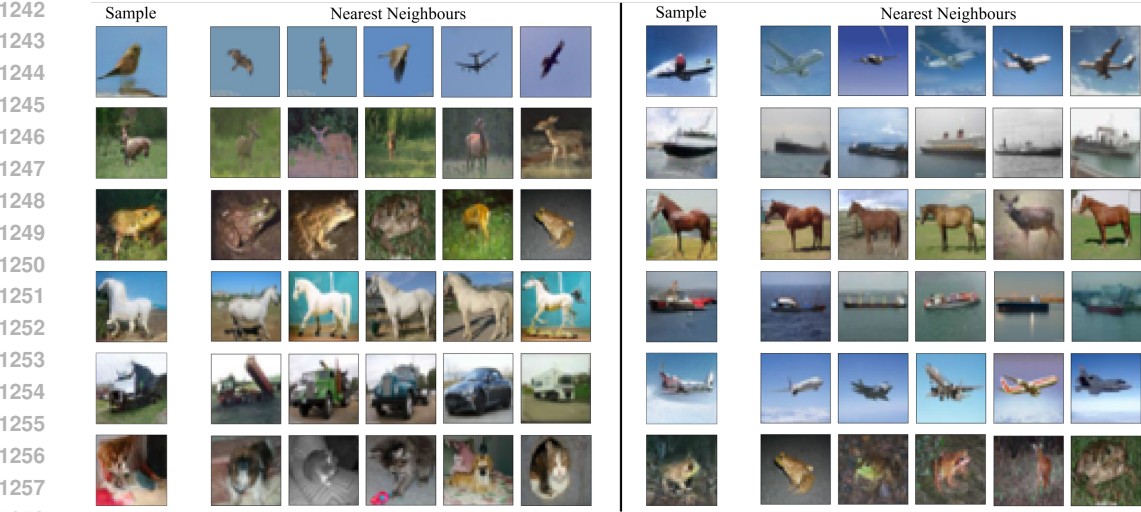

Figure 5: Generated samples and their five nearest neighbours in the CIFAR-10 train set based on pixel distance.

Table 3 compares the AUROC scores of VAPO-A with various likelihood-based and EBM-based models. The result shows that the model performs exceptionally well on the CIFAR-10 interpolated dataset. However, its performance is average on CIFAR-100 and SVHN. This suggests that the perturbation of training data using the data likelihood homotopy may not sufficiently explore the data space in comparison to MCMC methods. The investigation into the underlying cause is left for future work.

# E    LONG-RUN SDE SAMPLING

Figure 6 illustrate the long-run ($t_{\text{end}} >> 1$) SDE sampling results. The results demonstrate that the approximate likelihood density $\bar{\rho}(x; t)$ converges well to a stationary distribution. Nevertheless, the image quality degrades at large timesteps ($t = 10$), with the background details vanishing for example. This result is consistent with Agoritsas et al. (2023) that shows Langevin MCMC achieved the best EBM sampling results at some finite timestep (non-convergent). On that account, we choose a terminal time $t_{\text{end}} = 1.625$ for the numerical SDE solver.

Table 3: Comparison of AUROC scores ↑ for OOD detection on several datasets.

| Models | CIFAR-10 interpolation | CIFAR-100 | SVHN |
|---|---|---|---|
| PixelCNN | 0.71 | 0.63 | 0.32 |
| GLOW | 0.51 | 0.55 | 0.24 |
| NVAE | 0.64 | 0.56 | 0.42 |
| EBM-IG | 0.70 | 0.50 | 0.63 |
| VAEBM | 0.70 | 0.62 | 0.83 |
| CLEL | 0.72 | 0.72 | 0.98 |
| DRL | - | 0.44 | 0.88 |
| VAPO-A (Ours) | 0.78 | 0.50 | 0.61 |

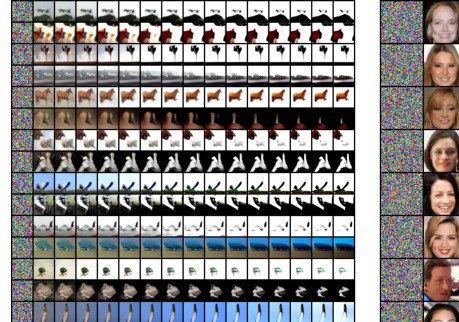 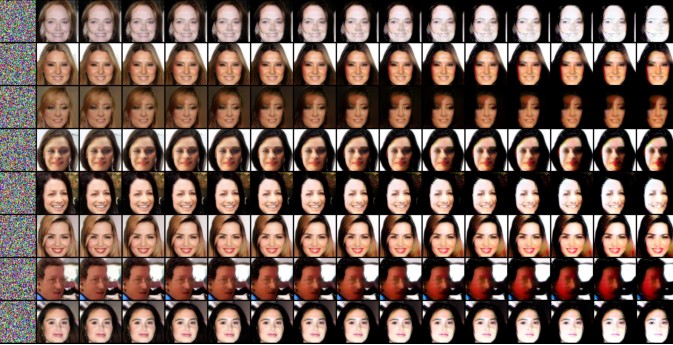

Figure 6: Long-run SDE sampling on time interval $t \in [0, 20]$ on unconditional CIFAR-10 $32 \times 32$ (left) and CelebA $64 \times 64$ (right). The samples converge well to their stationary states, albeit with image quality degrading at large timesteps due to oversaturation and loss of background detail, consistent with the results of Agoritsas et al. (2023).

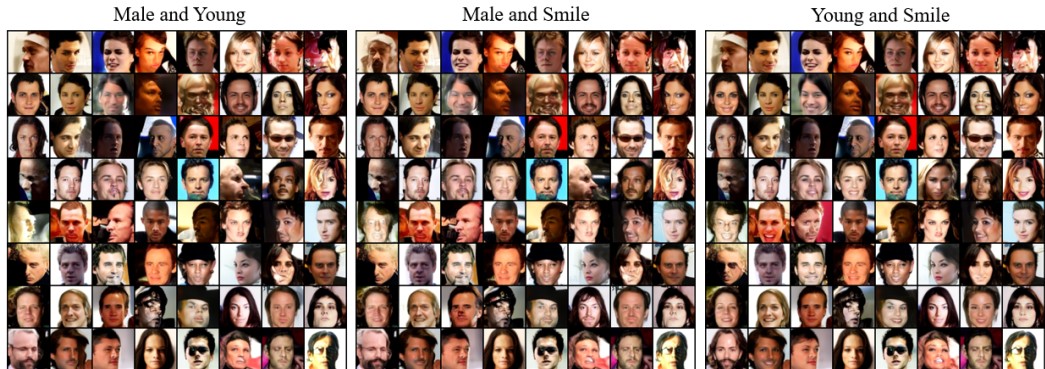

Figure 7: Compostional generation results on CelebA ($64 \times 64$)

## F  COMPOSITIONAL GENERATION

We assess the efficacy of the VAPO-T model in compositional generation by conducting experiments on the CelebA ($64 \times 64$) dataset with three attributes *Male, Smile*, and *Young* as the conditional concepts. Specifically, we aim to show the compositionality with the following attribute combinations: (*Male & Young*), (*Male & Smile*), and (*Young & Smile*). We train a class-conditional VAPO-T energy model $\Phi_\theta(x(t), c)$ based on classifier-free guidance Ho & Salimans (2021) to enable conditional generation. Subsequently, we estimate the energy of each conditional concept (attributes) individually, and take the normalized sum of the conditional energies $\sum_i w_i \, \Phi_\theta(x(t), c_i)$ where $c$ is the the conditioning attribute labels and $w$ is the composition weight with $\sum_i w_i = 1$. The normalized sum is then used to generate the samples. Here, we consider a composition between two CelebA attributes using equal weights.

## G  ADDITIONAL RESULTS

Figures 8 and 9 show additional examples of image interpolation on CIFAR-10 and CelebA $64 \times 64$, respectively. Figures 10 and 11 show additional uncurated examples of unconditional image generation on CIFAR-10 and CelebA $64 \times 64$, respectively.

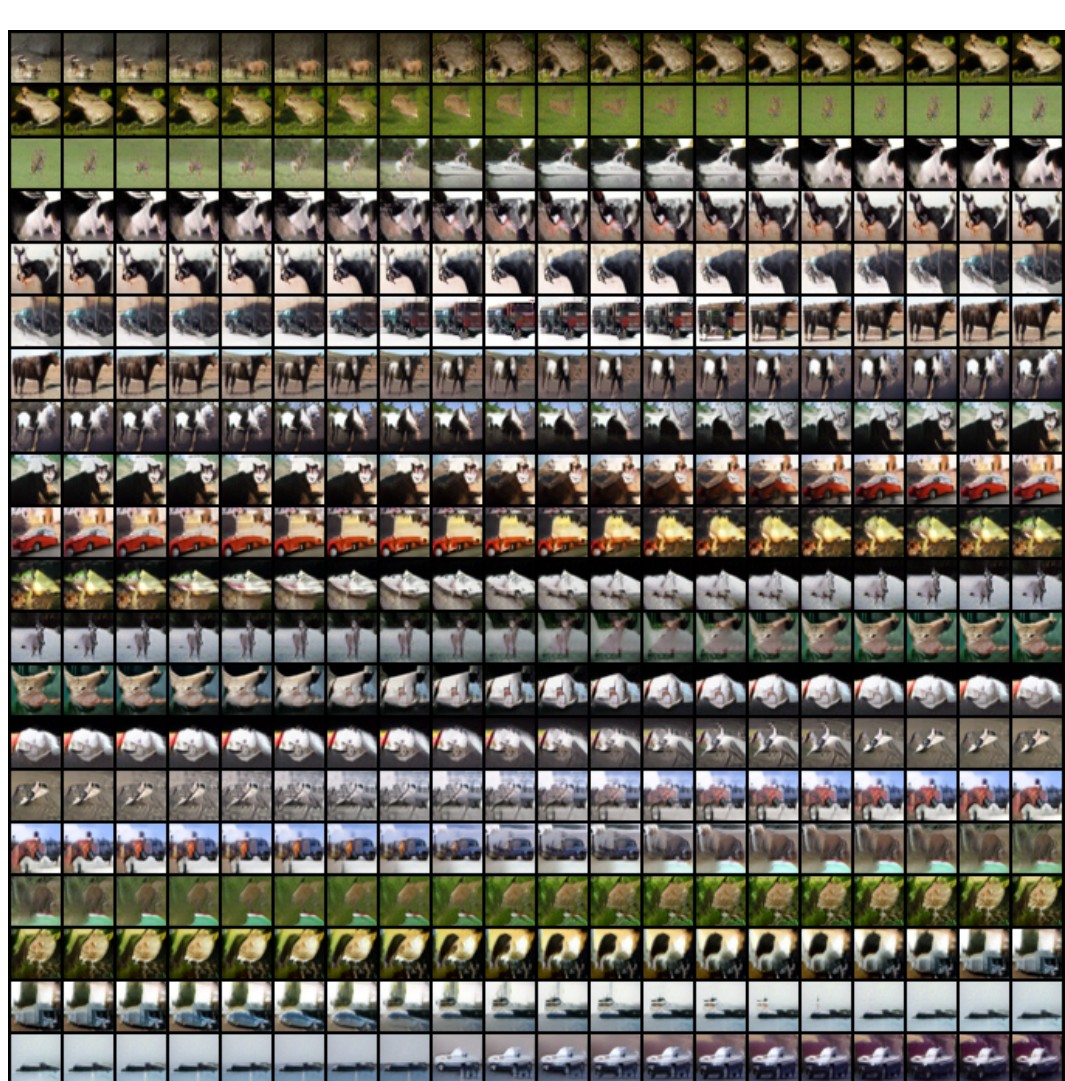

Figure 8: Additional interpolation results on unconditional CelebA $64 \times 64$.

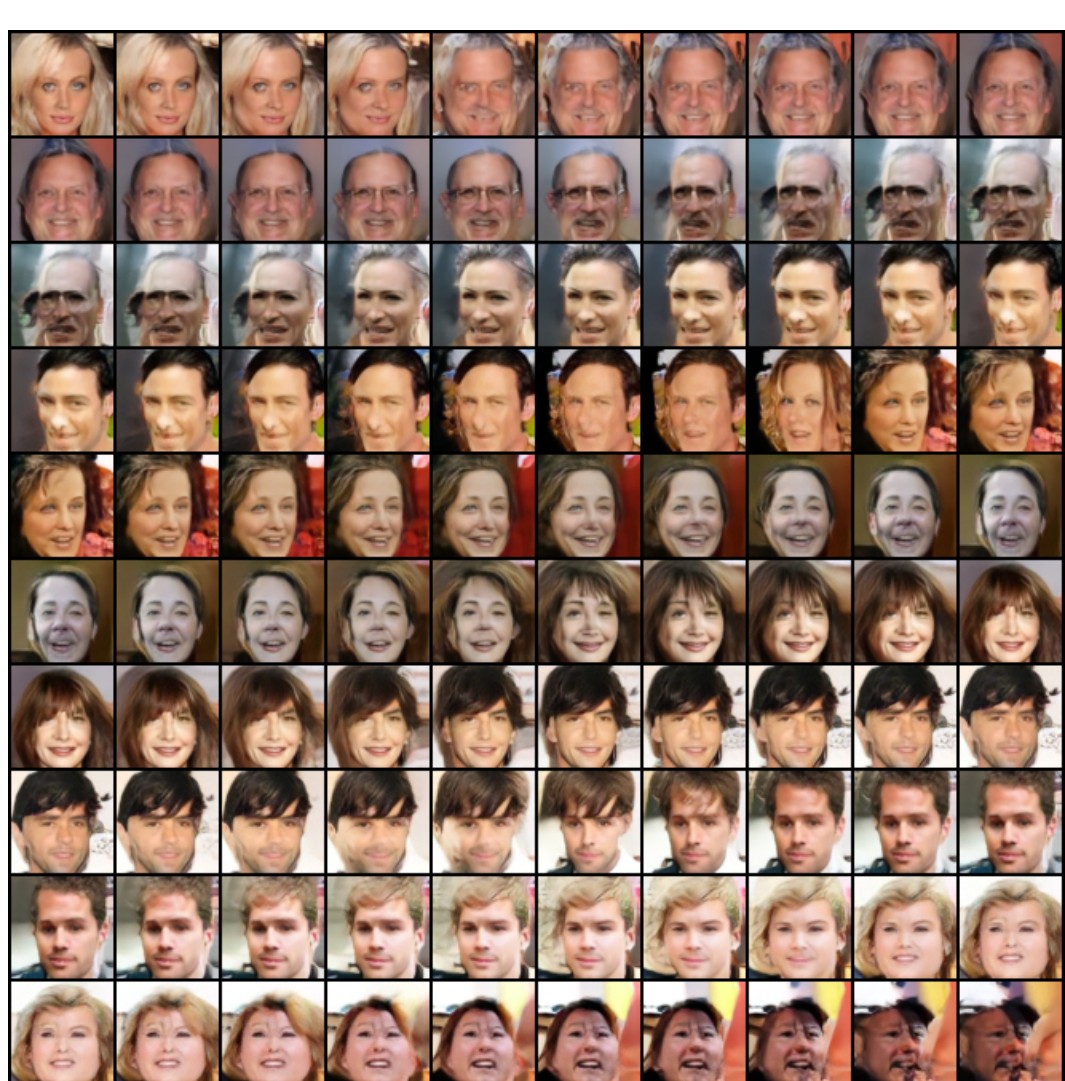

Figure 9: Additional interpolation results on unconditional CelebA $64 \times 64$.

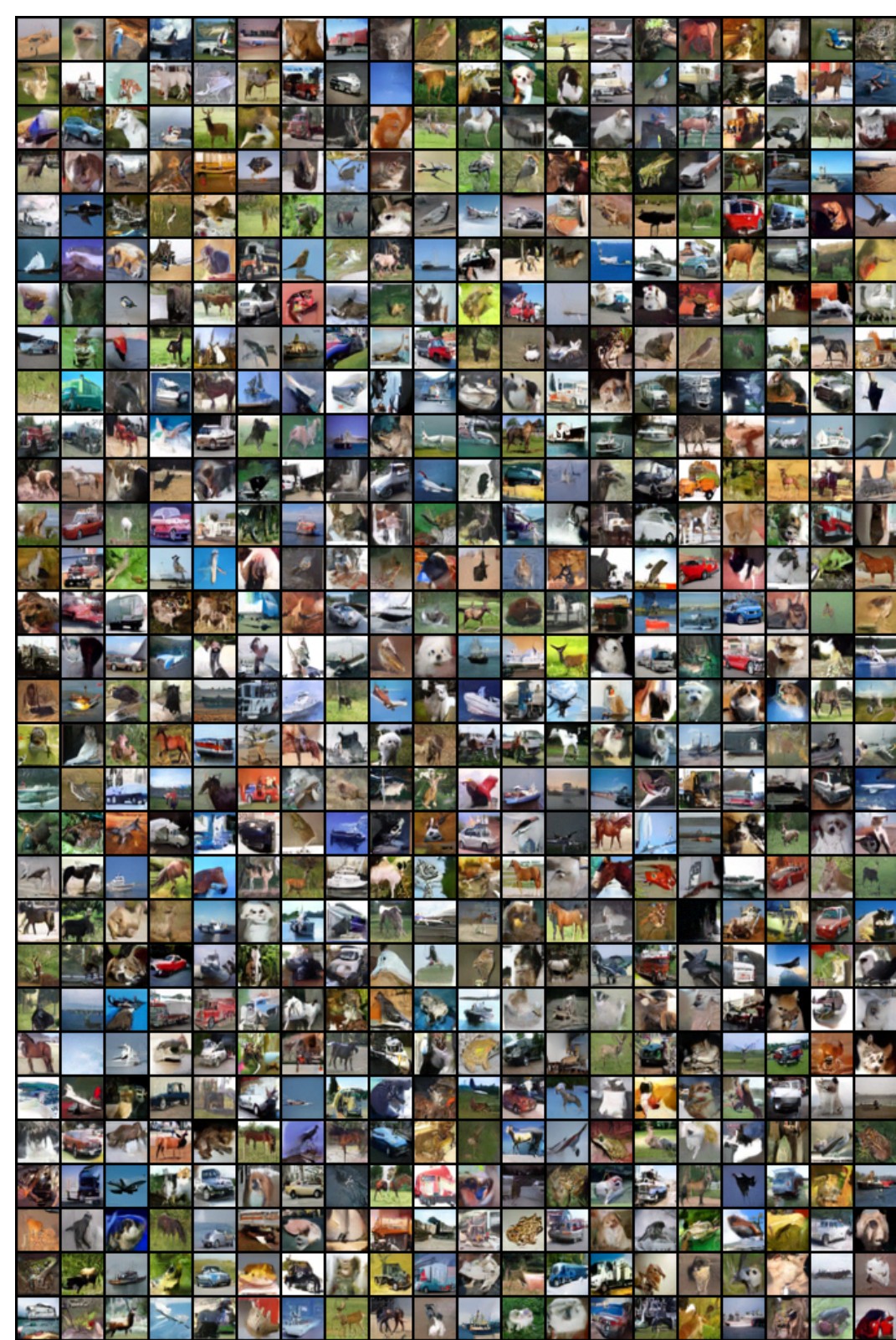

Figure 10: Additional uncurated samples on unconditional CIFAR-10 $32 \times 32$.

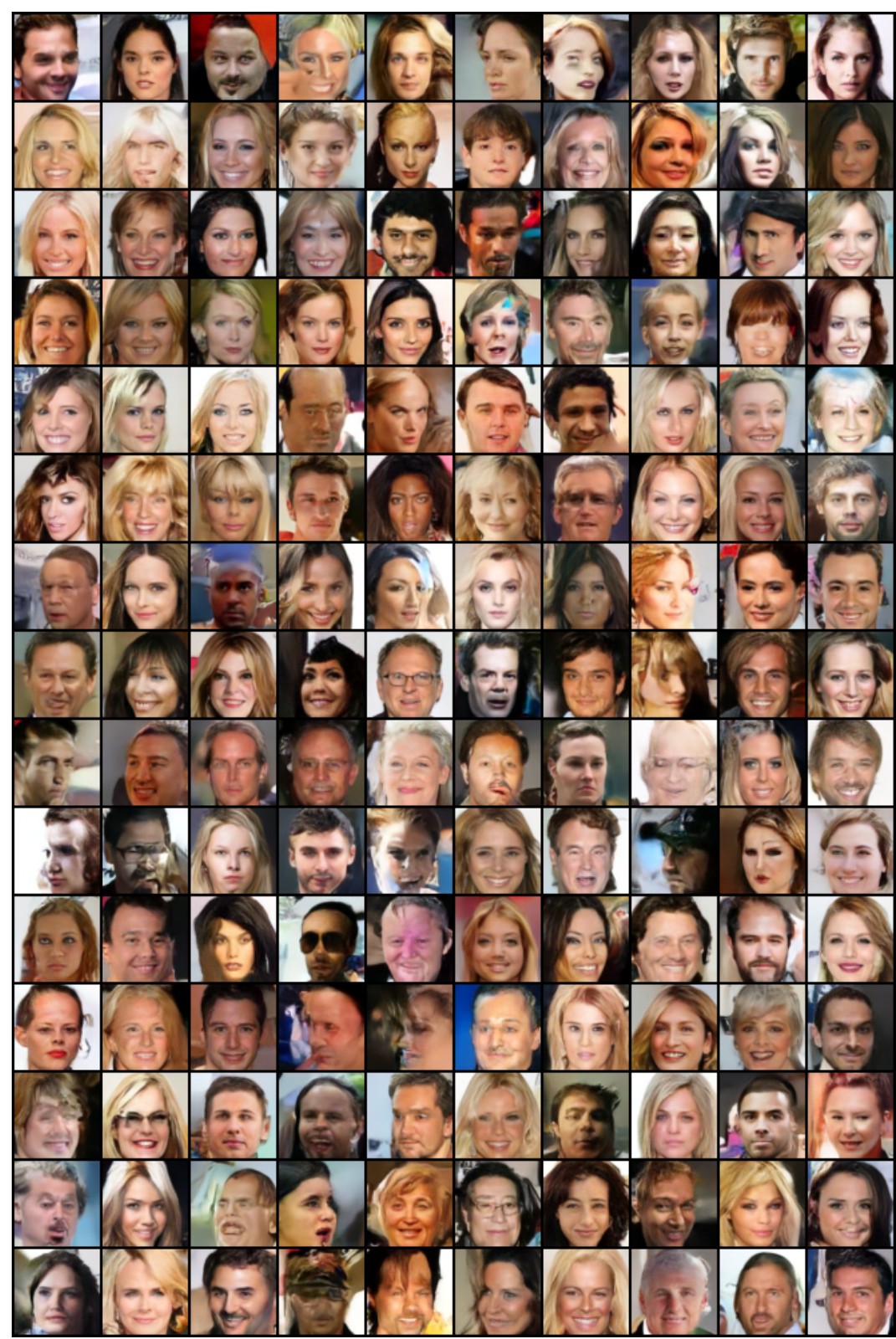

Figure 11: Additional uncurated samples on unconditional CelebA $64 \times 64$.

