# OpenReview forum: "Variational Potential Flow: A Novel Probabilistic Framework for Energy-Based Generative Modelling"
_ICLR.cc/2025/Conference — ICLR 2025 Conference Withdrawn Submission_

### Official Review · Reviewer_wVir · 2024-10-26

**Soundness:** 3
**Presentation:** 2
**Contribution:** 2
**Rating:** 3
**Confidence:** 4

**Summary:**

This paper introduces a new generative model framework called VAPO. This framework learns a potential energy function to guide prior samples through a density evolution that approximates the data likelihood, bypassing the need for implicit and unstable MCMC sampling. VAPO applies Poisson's equation and the deep Ritz method to define and solve the flow of prior samples, ensuring they align with the data likelihood homotopy. VAPO is tested on image datasets like CIFAR-10 and CelebA, achieving competitive FID scores for unconditional image generation compared to state-of-the-art models.

**Strengths:**

1. The VAPO method uses a potential energy function to guide the flow of prior samples toward the data likelihood, which is a fresh approach combining elements from particle flow and the Deep Ritz method. Experiments on CIFAR-10 and CelebA show that VAPO achieves competitive performance on FID scores relative to existing EBM-based approaches.

**Weaknesses:**

1. VAPO-A adopts the architecture from VAEBM [1], yet its FID scores fall short of baseline performance. This raises concerns regarding the overall advantage of VAPO. If the authors aim to address the high variance training, computational complexity, and low flexibility of MCMC and strengthen the claim of stability and efficiency of VAPO, the authors should conduct targeted experiments that compare training variance or computation time with MCMC-based methods. For example, record the loss value for each training epoch and calculate the standard deviation, measure the average time taken to generate a single sample, and record the total time and memory consumption required for the model to converge and compare it with MCMC methods. This would highlight the practical benefits of VAPO beyond standard performance metrics like FID.

[1] Zhisheng Xiao, Karsten Kreis, Jan Kautz, and Arash Vahdat. {VAEBM}: A symbiosis between variational autoencoders and energy-based models. In International Conference on Learning Representations, 2021.

2. VAPO incorporates computationally costly methods, such as numerical SDE solver. The authors also acknowledge that VAPO requires a large number of training iterations to converge, which might be a significant drawback for scaling this model to more complex or higher-dimensional datasets. While the elimination of MCMC improves training stability, the optimization process remains computationally expensive.

3. Previous approaches using MCMC methods and those based on flow models should be discussed together in the Related Work section and clearly referenced in the main text. This would provide readers with a comprehensive context for understanding the advancements presented in VAPO. Additionally, Algorithm 1, which outlines the algorithmic workflow, should be included in the main body of the paper rather than placed in the appendix. This is a critical component for grasping the methodology and overall contributions of the work. I recommend placing the Related Work section as Section 2 and positioning Algorithm 1 before Section 4 to enhance the flow and accessibility of the content.

4. Some sections, particularly those involving complex mathematical derivations, such as Sections 3.2 and 3.3, are difficult to follow due to the density of the technical content. These parts could be made more accessible by including additional explanatory text or visual aids to help guide readers through the more intricate aspects of the methodology. Moreover, the sketch proof of the main theorem, which is essential for understanding the core contributions, should be included in the main text.

5. The citation formatting throughout the paper is often incorrect. For instance, many references are not properly enclosed in brackets, which disrupts the flow of reading and makes it difficult to distinguish between the main text and references.

**Questions:**

1. What assumptions are made about the potential flow in the theoretical analysis? For example, are there any assumptions about the target distribution? Are these assumptions too restrictive to be realistically met in experiments, or how are they satisfied in practice?

2. In Line 199, what is $\bar{p}(x)$ and why is it stated that $\bar{p}(x)$ acts as a continuous interpolation of the data likelihood $p_{\text{data}}(x)$? Could you clarify what is meant by "continuous interpolation" in this context and how $\bar{p}(x)$ relates to $p_{\text{data}}(x)$?

---

### Official Review · Reviewer_LWEK · 2024-11-03

**Soundness:** 3
**Presentation:** 3
**Contribution:** 3
**Rating:** 6
**Confidence:** 4

**Summary:**

This paper introduces VAPO, a new framework to train Energy-Based Model(EBM). Different from traditional EBM training algorithms. VAPO does not require expensive MCMC sampling at each training step. Instead, VAPO defines an interpolation path $p(x, t)$ between a noise distribution $q(x)$ at $t=0$ and distribution $\bar{p}(x)$ at $t=1$. $\bar{p}(x)$ is the kernel density approximation of the true data likelihood. The authors then introduce a potential function $\Phi(x_t)$, whose gradient guides the transport equation of the defined density path. The potential function is learned by constructing a variational formulation of the homotopy path-matching problem. After the model is learned, $\nabla\Phi(x_t)$ is used to guide the data transformation in ODE during the test time to generate new samples. Experimental results show that VAPO achieves competitive generative performance among EBM baselines.

**Strengths:**

1. The paper proposes a novel framework for learning EBM. VAPO has the advantage of more efficient training without the needs of expensive MCMC sampling.
2. The paper provides a sound and detailed derivation and formulation. I appreciate its theoretical contribution.
3. The generative performance of VAPO is competitive among the EBM baselines.

**Weaknesses:**

My primary concern lies in the validity of the learned energy (or potential) function $\Phi(x_t)$. While the generative capability highlights one aspect of Energy-Based Models (EBMs), the ability to perform accurate density estimation is equally crucial. Although I acknowledge the competitive generative performance of VAPO, the relationship between the proposed potential function $\Phi(x_t)$ and the log data density remains unclear to me. Specifically:

1. Theoretically, what is the interpretation of the learned $\Phi(x_t)$? Are there any proofs ensuring that $\Phi(x_t)$ converges to the log data density? The paper demonstrates that the learned potential function $\Phi(x_t)$ can guide an ODE to generate valid samples via its gradient. However, a correct gradient field does not necessarily imply a valid energy function.

2. Experimentally, most of the results in the paper focus on demonstrating generative performance. I recommend that the authors include additional evaluations to demonstrate the validity of the learned energy function..

**Questions:**

Please check the weakness part.

---

### Official Review · Reviewer_EfBV · 2024-11-04

**Soundness:** 3
**Presentation:** 3
**Contribution:** 3
**Rating:** 6
**Confidence:** 4

**Summary:**

This work presents a novel framework for learning energy-based SDEs that can transport samples from a latent prior distribution to a distribution that closely approximates data samples (up to a small additive noise). First, a conditional density homotopy between the prior distribution of noise and the posterior distribution of a slightly perturbed sample given a data sample is defined, along with an unconditional homotopy defined by taking the product of the data density and conditional homotopy and integrating w.r.t. the data variable. This defines a continuous sequence of densities interpolating between the prior and approximate data distribution which will be the target of the generative model. The generative model is an energy function whose gradients are trained to act as the drift term of an SDE with predetermined diffusion coefficients. The evolution of this SDE is intended to match the density homotopy, so that the gradients of the energy function can produce samples along the homotopy trajectory that start from the prior and end at the approximate data distribution. To achieve this, several propositions are used to present analytical forms for the evolution of the homotopy over time, establish a Poisson equation which is equivalent to minimizing the KL-divergence between the homotopy densities and the optimal trained model, and establish a practical loss function that can enforce the Poisson equation and be used to train a neural network. Experiments are conducted for unconditional image generation on CIFAR-10 and Celeb-A which show strong performance among EBMs.

**Strengths:**

* This work brings an interesting and novel perspective to the study of generative modeling with potential energy. The density homotopy introduced in Section 3.1 is a fresh and potentially fruitful direction for using smooth interpolation between unnormalized densities in a manner similar to interpolation between data and prior samples in diffusion models and flow matching. Using this homotopy as the training target for a SDE with a learnable drift potential seems like a natural and effective choice.
* The theoretical background has a high degree of technical merit and thorough presentation. It is notable (and somewhat surprising) that the time evolution of the density homotopy has the simple and intuitive form in (8) and that the objective in (15) can be used to train a potential that can be used to match the homotopy over time. This could inspire future research in a similar direction.
* The experimental results show strong performance among EBM methods.

**Weaknesses:**

* There are a few hyperparameters which are not fully explored, such as the noise schedule $\beta (t)$ (left constant), perturbation level $\sigma$, spectral gap $\lambda$, and $\varepsilon$. Some discussion of why these were chosen and the sensitivity to these hyperparameters would be helpful. In particular, the $\beta$ schedule is crucial for diffusion models. Why is it not essential here? Could the results be improved with a better schedule?
* Although experimental results are strong among EBMs, they lag behind the diffusion and other SOTA methods. Furthermore, this work also performs slightly worse than the related work [a] which learns a potential energy and uses the SDE (10) to draw samples, which is based on a straightforward diffusion objective (which is actually a component of the proposed loss). In general, the paper would be strengthened by comparing the results from the proposed method with a more straightforward diffusion model using a potential energy network instead of a score network (such as [a]), since both cases are essentially training a network so that (10) transports samples from a latent prior to data.

[a] https://openreview.net/forum?id=9AS-TF2jRNb

**Questions:**

* Could the questions about hyperparameters in the first weakness be addressed?
* Could the questions about the relative performance of the proposed method and diffusion models parameterized by potential energy functions be addressed?

Technical:
* About the equation (6): if I am understanding correctly, joint probability of $x$ and $\bar{x}$ satisfies: $p( \bar{x} | x) q(x) = p(\bar{x}, x) = p (x | \bar{x}) p_\text{data} (\bar{x})$. But why does $p( \bar{x} | x) q(x) = p (x | \bar{x}) p_\text{data} (\bar{x})$?
* In (41) second and third equality, the signs be reversed after integration by parts, right?
* I am not sure how (46) and (47) imply $C=0$. It looks like it should instead be: $C = - (1/2) \int E [ \rho(x; \bar{x} (\gamma (x, \bar{x}) - \bar{\gamma} (x, \bar{x}))] dx$.

---

### Official Review · Reviewer_27rP · 2024-11-08

**Soundness:** 1
**Presentation:** 1
**Contribution:** 2
**Rating:** 3
**Confidence:** 3

**Summary:**

The paper introduces a new framework for energy-based generative models called Variational Potential Flow (VAPO). This approach eliminates the need for implicit MCMC sampling and does not depend on auxiliary latent models or cooperative training methods. The VAPO framework achieves this by learning a potential energy function path, where the gradient flow guides prior samples along an approximate data likelihood homotopy. Additionally, the authors develop an energy loss function through a variational formulation that leverages the KL divergence between the density evolution of the flow-driven prior and the data likelihood homotopy. The framework is tested on CIFAR-10 and CelebA datasets for unconditional image generation.

**Strengths:**

The paper aims to tackle an important problem that is reduce the complexity of energy-based generative models due to the reliance on MCMC sampling or auxiliary latent models.

**Weaknesses:**

-	The presentation could be improved, as the authors included extensive mathematical content in the main paper. The reviewer recommends simplifying this mathematical material within the main text and focusing on presenting high-level concepts and core results to enhance accessibility for readers.
-	The paper allocates considerable space to unnecessary mathematical content, which detracts from the clarity and quality of the experimental section. Additionally, the proposed method underperforms compared to many other energy-based model (EBM) approaches. Given that the paper's objective is to simplify and stabilize EBM components, such as MCMC sampling and auxiliary latent models, the authors should include a comprehensive analysis. This should involve demonstrating aspects like training convergence and complexity to better support the method's effectiveness and contributions.
-	In its current form, the paper presents extensive mathematical content along with some experimental results, but lacks thorough analysis. Consequently, it is unclear how beneficial the proposed tools truly are. The authors should consider demonstrating the proposed framework on simple, synthetic datasets to provide clearer insights into its behavior and effectiveness.

**Questions:**

-	The reviewer observed from the appendix that the authors used a lot of tricks for the network architecture and optimizer. Is this fair compared to the baselines presented in Table 1.

---

### Note · Authors · 2024-11-24

I have read and agree with the venue's withdrawal policy on behalf of myself and my co-authors.